# DART: A Principled Approach to Adversarially Robust Unsupervised Domain Adaptation

## Abstract

Distribution shifts and adversarial examples are two major challenges for deploying machine learning models. While these challenges have been studied individually, their combination is an important topic that remains relatively underexplored. In this work, we study the problem of adversarial robustness under a common setting of distribution shift – unsupervised domain adaptation (UDA). Specifically, given a labeled source domain $\mathcal{D}_S$ and an unlabeled target domain $\mathcal{D}_T$ with related but different distributions, the goal is to obtain an adversarially robust model for $\mathcal{D}_T$. The absence of target domain labels poses a unique challenge, as conventional adversarial robustness defenses cannot be directly applied to $\mathcal{D}_T$. To address this challenge, we first establish a generalization bound for the adversarial target loss, which consists of (i) terms related to the loss on the data, and (ii) a measure of worst-case domain divergence. Motivated by this bound, we develop a novel unified defense framework called *Divergence Aware adveRsarial Training* (DART), which can be used in conjunction with a variety of standard UDA methods; e.g., DANN (Ganin & Lempitsky, 2015). DART is applicable to general threat models, including the popular $\ell_p$-norm model, and does not require heuristic regularizers or architectural changes. We also release DomainRobust: a testbed for evaluating robustness of UDA models to adversarial attacks. DomainRobust consists of 4 multi-domain benchmark datasets (with 46 source-target pairs) and 7 meta-algorithms with a total of 11 variants. Our large-scale experiments demonstrate that on average, DART significantly enhances model robustness on all benchmarks compared to the state of the art, while maintaining competitive standard accuracy. The relative improvement in robustness from DART reaches up to 29.2% on the source-target domain pairs considered.

## 1 Introduction

In many machine learning applications, only unlabeled data is available and the cost of labeling can be prohibitive. In such cases, it is often possible to obtain labeled training data from a related *source domain*, which has a different distribution from the *target domain* (i.e., test-time data). As an example, suppose the target domain of interest consists of real photographs of objects. One appropriate source domain could be hand-drawn images of the same objects. Due to the distribution shift, learning models using only source data may lead to poor performance (Ganin & Lempitsky, 2015). To overcome this challenge, there has been extensive research on unsupervised domain adaptation (UDA) methods (Ben-David et al., 2006; Mansour et al., 2008; 2009; Wilson & Cook, 2020; Liu et al., 2022). Given labeled data from the source domain and only unlabeled data from the target domain, UDA methods aim to learn models that are robust to distribution shifts and that work well on the target domain.

While standard UDA methods have proven successful in various applications (Ghafoorian et al., 2017; Liu et al., 2021), they do not take into account robustness to adversarial attacks. These attacks involve carefully designed input perturbations that may deceive machine learning models (Szegedy et al., 2013; Goodfellow et al., 2014; Chakraborty et al., 2018; Hendrycks & Dietterich, 2019). The lack of adversarial robustness can be a serious obstacle for deploying models in safety-critical applications. A significant body of research has studied defense mechanisms for making models robust against adversarial attacks (Chakraborty et al., 2018; Ren et al., 2020). However, standard defenses are not designed to handle general distribution shifts, such as those in UDA. Specifically,

defenses applied on one domain may not generally transfer well to other domains. Thus, adversarial robustness is a major challenge for UDA.

In this work, we study the problem of adversarial robustness in UDA, from both theoretical and practical perspectives. Given labeled data from a source domain $\mathcal{D}_S$ and unlabeled data from a related target domain $\mathcal{D}_T$, our goal is to train a model that performs effectively on $\mathcal{D}_T$ while ensuring robustness against adversarial attacks. This requires controlling the *adversarial target loss* (i.e., the loss of the model on adversarial target examples), which cannot be computed directly due to the absence of target labels. To make the problem more tractable, we establish a new generalization bound on the target adversarial loss, which allows for upper bounding this loss by quantities that can be directly controlled. Motivated by the theory, we introduce DART, a unified defense framework against adversarial attacks, which can be used with a wide class of UDA methods and for general threat models. Through extensive experiments, we find that DART outperforms the state of the art on various benchmarks. Our contributions can be summarized as follows:

1. **Generalization Bound.** We establish a new generalization bound for the adversarial target loss. The bound consists of three quantities: the source domain loss, a measure of "worst-case" domain divergence, and the loss of an ideal classifier over the source domain and the "worst-case" target domain.

2. **Unified Defense Framework.** Building on our theory, we introduce **D**ivergence **A**ware adversa**R**ial **T**raining (DART), a versatile defense framework that can be used in conjunction with a wide range of distance-based UDA methods (e.g., DANN (Ganin & Lempitsky, 2015), MMD (Gretton et al., 2012), CORAL (Sun & Saenko, 2016), etc.). Our defenses are principled, apply to general threat models (including the popular $\ell_p$-norm threat model) and do not require specific architectural modifications.

3. **Testbed.** To encourage reproducible research in this area, we release DomainRobust[1], a testbed designed for evaluating the adversarial robustness of UDA methods, under the common $\ell_p$-norm threat model. DomainRobust consists of four multi-domain benchmark datasets: DIGITs (including MNIST, MNIST-M, SVHN, SYN, USPS), OfficeHome, PACS, and VisDA. DomainRobust encompasses seven meta-algorithms with a total of 11 variants, including DART, Adversarial Training (Madry et al., 2017), TRADES (Zhang et al., 2019a), and several recent heuristics for robust UDA such as ARTUDA (Lo & Patel, 2022) and SRoUDA (Zhu et al., 2023). The testbed is written in PyTorch and can be easily extended with new methods.

4. **Empirical Evaluations.** We conduct extensive experiments on DomainRobust under a white-box setting for all possible source-target dataset pairs. The results demonstrate that DART achieves better robust accuracy than the state-of-the-art on all 4 benchmarks considered, while maintaining competitive standard (a.k.a. clean) accuracy. For example, the average relative improvement across all 20 source-target domain pairs of DIGITs exceeds 5.5%, while the relative improvement of robust accuracy on individual source-target pairs reaches up to 29.2%.

## 1.1 RELATED WORK

**UDA.** In their seminal study, Ben-David et al. (2006) established generalization bounds for UDA, which were later extended and studied by various works (Mansour et al., 2009; Ben-David et al., 2010; Zhang et al., 2019b; Acuna et al., 2021); see Redko et al. (2020) for a survey of theoretical results. One fundamental class of practical UDA methods is directly motivated by these theoretical bounds and is known as Domain Invariant Representation Learning (DIRL). Popular DIRL methods work by minimizing two objectives: (i) empirical risk on the labeled source data, and (ii) some discrepancy measure between the feature representations of the source and target domain, making these representations domain invariant; e.g., DAN (Long et al., 2015), DANN (Ganin et al., 2016), CORAL (Sun & Saenko, 2016), MCD (Saito et al., 2018). However, both the theoretical results and practical UDA methods do not take adversarial robustness into consideration.

**Adversarial Robustness.** Understanding the vulnerability of deep models against adversarial examples is a crucial area of research (Akhtar & Mian, 2018; Zhang et al., 2020; Bai et al., 2021). Learning a classifier that is robust to adversarial attacks can be naturally cast as a robust (min-max) optimization problem (Madry et al., 2017). This problem can be solved using *adversarial training*: training the model over adversarial examples generated using constrained optimization algorithms

---

[1]Code will be released upon acceptance.

such as projected gradient descent (PGD). Unfortunately, adversarial training and its variants (e.g., TRADES (Zhang et al., 2019a), MART (Wang et al., 2019)) require labeled data from the target domain, which is unavailable in UDA. Another related line of work explores the transferability of robustness between domains (Shafahi et al., 2019), which still requires labeled target data to fine-tune the model.

**Adversarial Robustness in UDA.** Unlike the supervised learning setting, there has been a limited number of works that study adversarial robustness in UDA, which we discuss next. RFA (Awais et al., 2021) employed external adversarially pretrained ImageNet models for extracting robust features. However, such pretrained robust models may not be available for the task at hand, and they are typically computationally expensive to pretrain from scratch. ASSUDA (Yang et al., 2021) designed adversarial self-supervised algorithms for image segmentation tasks, with a focus on black-box attacks. Similarly, ARTUDA (Lo & Patel, 2022) proposed a self-supervised adversarial training approach, which entails using three regularizers and can be regarded as a combination of DANN (Ganin & Lempitsky, 2015) and TRADES (Zhang et al., 2019a). SRoUDA (Zhu et al., 2023) introduced data augmentation techniques to encourage robustness, alternating between a meta-learning step to generate pseudo labels for the target and an adversarial training step (based on pseudo labels). While all these algorithms demonstrated promising results, they are heuristic in nature. In contrast, our algorithm DART is not only theoretically justified but also exhibits excellent performance–it outperforms ARTUDA and SRoUDA on all the benchmarks considered.

## 2 PROBLEM SETUP AND PRELIMINARIES

In this section, we formalize the problem setup and introduce some preliminaries on UDA theory.

**UDA setup.** Without loss of generality, we focus on binary classification with an input space $\mathcal{X} \subseteq \mathbb{R}^d$ (e.g., space of images) and an output space $\mathcal{Y} = \{\pm 1\}$. Let $\mathcal{H} \subseteq \{h : \mathcal{X} \to \mathcal{Y}\}$ be the hypothesis class and denote the loss function by $\ell : \mathbb{R} \times \mathcal{Y} \to \mathbb{R}_+$. We define the source domain $\mathcal{D}_S$ and target domain $\mathcal{D}_T$ as probability distributions over $\mathcal{X} \times \mathcal{Y}$. Given an arbitrary distribution $\mathcal{D}$ over $\mathcal{X} \times \mathcal{Y}$, we use the notation $\mathcal{D}^X$ to refer to the marginal distribution over $\mathcal{X}$; e.g., $\mathcal{D}_T^X$ denotes the unlabeled target domain. During training, we assume that the learner has access to a labeled source dataset $\mathcal{Z}_S = \{(x_i^s, y_i^s)\}_{i=1}^{n_s}$ drawn i.i.d. from $\mathcal{D}_S$ and an unlabeled target dataset $\{x_i^t\}_{i=1}^{n_t}$ drawn i.i.d. from $\mathcal{D}_T^X$. We use $X_S$ and $X_T$ to refer to the $n_s \times d$ source data matrix and $n_t \times d$ target data matrix, respectively.

**Robustness setup.** We assume a general threat model where the adversary's perturbation set is denoted by $\mathcal{B} : \mathcal{X} \to 2^{\mathcal{X}}$. Specifically, given an input example $x \in \mathcal{X}$, $\mathcal{B}(x) \subseteq \mathbb{R}^d$ represents the set of possible perturbations of x that an adversary can choose from. One popular example is the standard $\ell_p$ threat model that adds imperceptible perturbations to the input: $\mathcal{B}(x) = \{\tilde{x} : \|\tilde{x} - x\|_p \leq \alpha\}$ for a fixed norm $p$ and a sufficiently small $\alpha$. In the context of image classification, another example of $\mathcal{B}(x)$ could be a discrete set of large-norm (perceptible) transformations such as blurring, weather corruptions, and image overlays (Hendrycks & Dietterich, 2019; Stimberg et al., 2023). In what follows, our theoretical results will be applicable to a general $\mathcal{B}(x)$, and our experiments will be based on the standard $\ell_p$ threat model.

We denote the standard loss and the adversarial loss of a classifier $h$ on a distribution $\mathcal{D}$ by

$$L(h; \mathcal{D}) := \mathbb{E}_{(x,y) \sim \mathcal{D}} [\ell(h(x), y)] \quad \text{and} \quad L_{\text{adv}}(h; \mathcal{D}) := \mathbb{E}_{(x,y) \sim \mathcal{D}} \sup_{\tilde{x} \in \mathcal{B}(x)} [\ell(h(\tilde{x}), y)],$$

respectively. Given source samples $\mathcal{Z}_S$, we denote the empirical standard source loss as $L(h; \mathcal{Z}_S) := \frac{1}{n} \sum_{i=1}^{n} \ell(h(x_i^s), y_i^s)$. We add superscript 0/1 when considering 0-1 loss; i.e, $\ell^{0/1}, L^{0/1}, L_{\text{adv}}^{0/1}$. Our ultimate goal is to find a robust classifier $h$ that performs well against adversarial perturbations on the target domain; i.e., $h = \arg\min_{h \in \mathcal{H}} L_{\text{adv}}^{0/1}(h; \mathcal{D}_T)$.

### 2.1 STANDARD UDA THEORY

In this section, we briefly review key quantities and a UDA learning bound that has been introduced in the seminal work of Ben-David et al. (2010) – these will be important for the generalization bound we introduce in Section 3. We first introduce $\mathcal{H}\Delta\mathcal{H}$-divergence, which measures the ability of the hypothesis class $\mathcal{H}$ to distinguish between samples from two input distributions.

**Definition 2.1** ($\mathcal{H}\Delta\mathcal{H}$-divergence (Ben-David et al., 2010)). Given some fixed hypothesis class $\mathcal{H}$, let $\mathcal{H}\Delta\mathcal{H}$ denote the symmetric difference hypothesis space, which is defined by: $h \in \mathcal{H}\Delta\mathcal{H} \Leftrightarrow h(\mathrm{x}) = h_1(\mathrm{x}) \oplus h_2(\mathrm{x})$ for some $(h_1, h_2) \in \mathcal{H}^2$, where $\oplus$ stands for the XOR operation. Let $\mathcal{D}_S^X$ and $\mathcal{D}_T^X$ be two distributions over $\mathcal{X}$. Then the $\mathcal{H}\Delta\mathcal{H}$-divergence between $\mathcal{D}_S^X$ and $\mathcal{D}_T^X$ is defined as:

$$d_{\mathcal{H}\Delta\mathcal{H}}(\mathcal{D}_S^X, \mathcal{D}_T^X) = 2 \sup_{h \in \mathcal{H}\Delta\mathcal{H}} \left| \mathbb{E}_{\mathrm{x}\sim\mathcal{D}_S^X} \mathbb{1}\left[h(\mathrm{x}) = 1\right] - \mathbb{E}_{\mathrm{x}\sim\mathcal{D}_T^X} \mathbb{1}\left[h(\mathrm{x}) = 1\right] \right|,$$

where $\mathbb{1}(\cdot)$ is the indicator function.

Here $d_{\mathcal{H}\Delta\mathcal{H}}(\mathcal{D}_S^X, \mathcal{D}_T^X)$ captures an interesting interplay between the hypothesis class and the source/target distributions. On one hand, when the two distributions are fixed, a richer $\mathcal{H}$ tends to result in a larger $\mathcal{H}\Delta\mathcal{H}$-divergence. On the other hand, for a fixed $\mathcal{H}$, greater dissimilarity between the two distributions leads to a larger $\mathcal{H}\Delta\mathcal{H}$-divergence. In practice, $\mathcal{H}\Delta\mathcal{H}$-divergence is generally intractable to compute exactly, but it can be approximated using finite samples, as we will discuss in later sections. With this definition, Ben-David et al. (2010) established an important upper bound on the standard target loss, which we recall in the following theorem.

**Theorem 2.1** (Ben-David et al. (2010)). Given a hypothesis class $\mathcal{H}$, the following holds:

$$\underbrace{L^{0/1}(h; \mathcal{D}_T)}_{\text{Target Loss}} \leq \underbrace{L^{0/1}(h; \mathcal{D}_S)}_{\text{Source Loss}} + \frac{1}{2} \underbrace{d_{\mathcal{H}\Delta\mathcal{H}}(\mathcal{D}_S^X, \mathcal{D}_T^X)}_{\text{Domain Divergence}} + \underbrace{\gamma(\mathcal{D}_S, \mathcal{D}_T)}_{\text{Ideal Joint Loss}}. \tag{1}$$

$\gamma(\mathcal{D}_S, \mathcal{D}_T) := \min_{h^* \in \mathcal{H}} \left[ L^{0/1}(h^*; \mathcal{D}_S) + L^{0/1}(h^*; \mathcal{D}_T) \right]$ is the joint loss of an ideal classifier that works well on both domains.

We note that Ben-David et al. (2010) also established a corresponding generalization bound, but the simpler bound above is sufficient for our discussion. The ideal joint loss $\gamma$ can be viewed as a measure of both the label agreement between the two domains and the richness of the hypothesis class, and it cannot be directly computed or controlled as it depends on the target labels (which are unavailable under UDA). If $\gamma$ is large, we do not expect a classifier trained on the source to perform well on the target, and therefore $\gamma$ is typically assumed to be small in the UDA literature. In fact, David et al. (2010) showed that having a small domain divergence and a small ideal joint risk is necessary and sufficient for transferability. Assuming a small $\gamma$, Theorem 2.1 suggests that the target loss can be controlled by ensuring that both the source loss and domain divergence terms in (1) are small – we revisit some practical algorithms for ensuring this in Section 4.

## 3 ADVERSARIALLY ROBUST UDA THEORY

In this section, we derive an upper bound on the adversarial target loss, which will be the basis of our proposed defense framework. We present our main theorem below and defer the proof to Appendix A.

**Theorem 3.1.** Let $\mathcal{H}$ be a hypothesis class with finite VC dimension $\text{VC}(\mathcal{H})$ and adversarial VC dimension $\text{AVC}(\mathcal{H})$ (Cullina et al., 2018). If $\mathcal{Z}_S$ and $\mathcal{Z}_T$ are labeled samples of size[2] $n$ drawn i.i.d. from $\mathcal{D}_S$ and $\mathcal{D}_T$, respectively, and $\mathrm{X}_S$ and $\mathrm{X}_T$ are the corresponding data matrices, then for any $\delta \in (0, 1)$, w.p. at least $1 - \delta$, for all $h \in \mathcal{H}$,

$$L_{\text{adv}}^{0/1}(h; \mathcal{D}_T) \leq \underbrace{L^{0/1}(h; \mathcal{Z}_S)}_{\text{Source Loss}} + \underbrace{\sup_{\substack{\tilde{\mathrm{x}}_i^t \in \mathcal{B}(\mathrm{x}_i^t), \forall i \in [n], \\ \tilde{\mathcal{Z}}_T = \left\{ (\tilde{\mathrm{x}}_i^t, y_i^t) \right\}_{i=1}^n}}}_{\text{Worst-case target}} \left[ \underbrace{d_{\mathcal{H}\Delta\mathcal{H}}(\mathrm{X}_S, \tilde{\mathrm{X}}_T)}_{\text{Domain Divergence}} + 2 \underbrace{\gamma(\mathcal{Z}_S, \tilde{\mathcal{Z}}_T)}_{\text{Ideal Joint Loss}} \right] + \epsilon, \tag{2}$$

where the generalization gap $\epsilon = \mathcal{O}(\sqrt{\frac{\max\{\text{VC}(\mathcal{H}), \text{AVC}(\mathcal{H})\} \log(n) + \log(1/\delta)}{n}})$, the (empirical) ideal joint loss is defined as $\gamma(\mathcal{Z}_S, \mathcal{Z}_T) := \min_{h^* \in \mathcal{H}} \left[ L^{0/1}(h^*; \mathcal{Z}_S) + L^{0/1}(h^*; \mathcal{Z}_T) \right]$, and the (empirical) $\mathcal{H}\Delta\mathcal{H}$-divergence can be computed as follows[3]:

$$d_{\mathcal{H}\Delta\mathcal{H}}(\mathrm{X}_S, \mathrm{X}_T) = 2\left( 1 - \min_{h \in \mathcal{H}\Delta\mathcal{H}} \left[ \frac{1}{n} \sum_{\mathrm{x}:h(\mathrm{x})=0} \mathbb{1}(\mathrm{x} \in \mathrm{X}_S) + \frac{1}{n} \sum_{\mathrm{x}:h(\mathrm{x})=1} \mathbb{1}(\mathrm{x} \in \mathrm{X}_T) \right] \right). \tag{3}$$

---

[2]We assume that $\mathcal{Z}_S$ and $\mathcal{Z}_T$ have the same size for simplicity. The result still applies to different sizes.

[3]In Definition 2.1, we defined $d_{\mathcal{H}\Delta\mathcal{H}}$ for two input distributions. Here we use an equivalent definition in which the two inputs are data matrices.

Theorem 3.1 states that the adversarial target loss can be bounded from above by three main terms (besides generalization error $\epsilon$): source loss, domain divergence, and the ideal joint loss. These three terms are similar to those in the bound of Theorem 2.1 for standard UDA; however, the main difference lies in that Theorem 3.1 evaluates the domain divergence and ideal joint loss terms for a "worst-case" target domain (instead of the original target domain). The first two terms (source loss and domain divergence) do not require target labels and can thus be directly computed and controlled. However, the ideal joint risk in Theorem 3.1 requires labels from the target domain and cannot be directly computed.

In Section 2.1, we discussed how the ideal joint loss in the standard UDA setting is commonly assumed to be small and is thus not controlled in many popular practical methods. Specifically, if we decompose $h$ into a feature extractor $g$ and a classifier $f$ (i.e., $h = f \circ g$), the ideal joint loss can be written as a function of $g$: $\gamma(\mathcal{D}_S, \mathcal{D}_T, g) := \min_{f^*: f^* \circ g \in \mathcal{H}} \left[ L^{0/1}(f^* \circ g; \mathcal{D}_S) + L^{0/1}(f^* \circ g; \mathcal{D}_T) \right]$. In the literature (Ben-David et al., 2006), $\gamma(\mathcal{D}_S, \mathcal{D}_T, g)$ is commonly assumed to be small for any reasonable $g$ that is chosen by the learning algorithm. However, for a fixed $g$, the ideal joint loss with the worst-case target in our setting may be generally larger than that of the standard UDA setting. While one possibility is to assume this term remains small (as in the standard UDA setting), we hypothesize that in practice it may be useful to control this term by finding an appropriate feature extractor $g$. In the next section, we discuss a practical defense framework that attempts to minimize the adversarial target risk by controlling all three terms in Theorem 3.1, including the ideal joint loss. In the experiments, we also present evidence that controlling all three terms typically leads to better results than controlling only the source loss and domain divergence.

## 4 Divergence Aware Adversarial Training: a practical defense

Recall that in the standard UDA setting, a fundamental class of UDA methods – DIRL – are based on the upper bound (1) or variants that use other domain divergence measures (Ganin et al., 2016; Li et al., 2017a; Zellinger et al., 2017). These methods are based on neural networks consisting of two main components: a feature extractor $g$ that generates feature representations and a classifier $f$ that generates the model predictions. Given an example x (from either the source or target domain), the final model prediction is given by $f(g(\mathrm{x}))$ (which we also write as $(f \circ g)(\mathrm{x})$, $h = f \circ g \in \mathcal{H}$). The key insight is that if the feature representations generated by $g$ are domain-invariant (i.e., they are similar for both domains), then the domain divergence will be small. Practical algorithms use a regularizer $\Omega$ that acts as a proxy for domain divergence. Thus, the upper bound on the standard target loss in (1) can be controlled by identifying a feature transformation $g$ and a classifier $f$ that minimize the combined effect of the source loss and domain divergence; i.e.,

$$\min_{g,f} \quad \underbrace{L(f \circ g; \mathcal{Z}_S)}_{\text{Empirical Source Loss}} + \underbrace{\Omega(\mathrm{X}_S, \mathrm{X}_T, g)}_{\text{Empirical Proxy for Domain Divergence}} \quad . \tag{4}$$

Such strategy is the basis behind several practical UDA methods, such as Domain Adversarial Neural Networks (DANN) (Ganin et al., 2016), Deep Adaptation Networks (Li et al., 2017a), and CORAL (Sun & Saenko, 2016). As an example, DANN directly approximates $\mathcal{H} \Delta \mathcal{H}$-divergence in (3); it defines $\Omega$ as the loss of a "domain classifier", which tries to distinguish between the examples of the two domains (based on the feature representations generated by $g$). We list some common UDA methods and their corresponding $\Omega$ in Appendix C.2.

We now propose a practical defense framework based on the theoretical guarantees that we derived in Section 3, namely DART (**D**ivergence **A**ware adve**R**sarial **T**raining).

**A practical bound.** We consider optimizing upper bound (2) in Theorem 3.1. Given a feature extractor $g$ and a classifier $f$, the upper bound in Theorem 3.1 can be rewritten as follows,

$$L_{\text{adv}}^{0/1}(f \circ g; \mathcal{D}_T) \leq L^{0/1}(f \circ g; \mathcal{Z}_S) + \sup_{\substack{\tilde{\mathrm{x}}_i^t \in \mathcal{B}(\mathrm{x}_i^t), \forall i \in [n] \\ \tilde{\mathcal{Z}}_T = \left\{(\tilde{\mathrm{x}}_i^t, y_i^t)\right\}_{i=1}^n}} \left[ d_{\mathcal{H} \Delta \mathcal{H}}(\mathrm{X}_S, \tilde{\mathrm{X}}_T) + 2\gamma(\mathcal{Z}_S, \tilde{\mathcal{Z}}_T, g) \right] + \epsilon, \tag{5}$$

Note that minimizing this bound requires optimizing both $g$ and $f$. Moreover, for any given $g$, the ideal joint loss $\gamma(\mathcal{Z}_S, \tilde{\mathcal{Z}}_T, g)$ requires optimizing a separate model. To avoid optimizing separate models at each iteration and obtain a more practical method, we further upper bound Equation (5). Specifically, we note that the ideal joint loss can be upper bounded as follows: $\gamma(\mathcal{Z}_S, \mathcal{Z}_T, g) = \min_{f^*: f^* \circ g \in \mathcal{H}} \left[ L^{0/1}(f^* \circ g; \mathcal{Z}_S) + L^{0/1}(f^* \circ g; \mathcal{Z}_T) \right] \leq (L^{0/1}(f \circ g; \mathcal{Z}_S) + L^{0/1}(f \circ g; \mathcal{Z}_T))$ for any $f$ such that $f \circ g \in \mathcal{H}$. Plugging the latter bound in Equation (2) gives us the following:

$$L_{\text{adv}}^{0/1}(f \circ g; \mathcal{D}_T) \le 3L^{0/1}(f \circ g; \mathcal{Z}_S) + \sup_{\tilde{x}_i^t \in \mathcal{B}(x_i^t), \forall i \in [n]} \left[ d_{\mathcal{H} \Delta \mathcal{H}}(X_S, \tilde{X}_T) + 2L^{0/1}(f \circ g; \tilde{\mathcal{Z}}_T) \right] + \epsilon. \quad (6)$$

**DART's optimization formulation.** DART is directly motivated by bound (6). To approximate the latter bound, we first fix some UDA method that satisfies form (4) and use the corresponding $\Omega$ as an approximation of $d_{\mathcal{H} \Delta \mathcal{H}}$. Let $\tilde{\mathcal{Z}}_S = \{(\tilde{x}_i^s, y_i^s)\}_{i=1}^{n_s}$ denote the source data (which can be either the original, clean source $\mathcal{Z}_S$ or potentially a transformed version of it, as we discuss later) and let $\tilde{X}_S$ be the corresponding data matrix. To approximate the third term in (6), we assume access to a vector of target pseudo-labels $\hat{Y}_T$ corresponding to the target data matrix $X_T$ – we will discuss how to obtain pseudo-labels later in this section. Using the latter approximations in bound (6), we train an adversarially robust classifier by solving the following optimization problem:

$$\min_{g,f} \quad L(f \circ g; \tilde{\mathcal{Z}}_S) + \sup_{\tilde{x}_i^t \in \mathcal{B}(x_i^t), \forall i \in [n_t]} \left[ \lambda_1 \Omega(\tilde{X}_S, \tilde{X}_T, g) + \lambda_2 L(f \circ g; (\tilde{X}_T, \hat{Y}_T)) \right], \quad (7)$$

where $(\lambda_1, \lambda_2)$ are tuning parameters. We remark that problem (7) represents a general optimization formulation–the choice of the optimization algorithm depends on the model $(g, f)$ as well as the nature of the perturbation set $\mathcal{B}$. If a neural network is used along with the standard $\ell_p$-norm perturbation set, then problem (7) can be optimized similar to standard adversarial training, i.e., the network can be optimized using gradient-based algorithms like SGD, and at each iteration the adversarial target examples $\tilde{X}_T$ can be generated via projected gradient descent (PGD) (Madry et al., 2017). In Appendix B.1, we present a concrete instance of framework (7) for the common $\ell_p$ threat model and using DANN as the base UDA method. We provide the pseudocode of DART in Appendix B.2.

**Pseudo-Labels $\hat{Y}_T$.** The third term in bound (7) requires target labels, which are unavailable under the UDA setup. We thus propose using pseudo-labels, which can be obtained through various methods. Here, we describe a simple approach that assumes access to a proxy for evaluating the model's accuracy (standard or robust) on the target domain. This is the same proxy used for hyper-parameter tuning. For example, this proxy could be the accuracy on a small, labeled validation set if available or any UDA model selection criterion (Wilson & Cook, 2020, Section 4.7). We maintain a pseudo-label predictor that aims at generating pseudo-labels for the target data. Initially, this predictor is pretrained using a standard UDA method in Equation (4). We then use these pseudo-labels to optimize the model $(g, f)$ as in (7). To improve the quality of the pseudo-labels, we periodically evaluate the model's performance (standard accuracy) based on the pre-selected proxy and assign the model weights to the pseudo-label predictor if the model performance has improved–see Appendix B.3 for details.

**Source Choices $\tilde{\mathcal{Z}}_S$.** We investigate three natural choices of transformations of the source data $\tilde{\mathcal{Z}}_S = \{(\tilde{x}_i^s, y_i^s)\}_{i=1}^{n_s}$: 1) Clean source: use the original (clean) source data; i.e., $\tilde{x}_i^s = x_i^s$. 2) Adversarial source: choose the source data that maximizes the adversarial source loss; i.e., $\tilde{x}_i^s = \arg\max_{\tilde{x}_i \in \mathcal{B}(x_i^s)} \ell(h(\tilde{x}_i); y_i^s)$, which is the standard way of generating adversarial examples. 3) KL source: choose the source data that maximizes the Kullback-Leibler (KL) divergence of the clean and adversarial predictions (Zhang et al., 2019a); i.e., $\tilde{x}_i^s = \arg\max_{\tilde{x}_i \in \mathcal{B}(x_i^s)} \text{KL}(h(\tilde{x}_i), h(x_i^s))$. At each iteration, the adversarial and KL sources can be generated using the same optimization algorithm used to generate the adversarial target examples (e.g., PGD for an $\ell_p$ perturbation set).

## 5 EMPIRICAL EVALUATION

### 5.1 DOMAINROBUST: A PYTORCH TESTBED FOR UDA UNDER ADVERSARIAL ATTACKS

We conduct large-scale experiments on DomainRobust: our proposed testbed for evaluating adversarial robustness under the UDA setting. DomainRobust focuses on image classification tasks, including 4 multi-domains meta-datasets and 11 algorithms. Our implementation is PyTorch-based and builds up on DomainBed (Gulrajani & Lopez-Paz, 2020), which was originally developed for evaluating the (standard) accuracy of domain generalization algorithms.

**Datasets.** DomainRobust includes four multi-domain meta-datasets: 1) DIGIT datasets (Peng et al., 2019) (includes 5 popular digit datasets across 10 classes, namely MNIST (LeCun et al., 1998), MNIST-M (Ganin & Lempitsky, 2015), SVHN (Netzer et al., 2011), SYN (Ganin & Lempitsky, 2015), USPS (Hull, 1994)); 2) OfficeHome (Venkateswara et al., 2017) (includes 4 domains across 65 classes: Art, Clipart, Product, RealWorld); 3) PACS (Li et al., 2017b) (includes 4 domains across 7 classes: Photo, Art Painting, Cartoon, SKetch); 4) VisDA (Peng et al., 2017) (includes 2 domains across 12 classes: Synthetic and Real). Further details of each dataset are presented in Appendix C.1. We consider all pairs of source and target domains for each dataset.

**Algorithms.** We study 7 meta-algorithms (with a total of 11 variants). Unless otherwise noted, we use DANN as the base UDA method, i.e., we fix the domain divergence $\Omega$ to be DANN's regularizer and use it for all algorithms (except source-only models). We consider the following algorithms:

- **Natural DANN**. This is standard DANN without any defense mechanism.
- Source-only models, which include **AT(src)** and **TRADES(src)**. We apply Adversarial Training (Madry et al., 2017) and TRADES (Zhang et al., 2019a) only on labeled source data.
- Pseudo-labeled target models, which include **AT(tgt,pseudo)** and **TRADES(tgt,pseudo)**. We first train a standard DANN and use it to predict pseudo-labels for the unlabeled target data. We then apply standard adversarial training or TRADES on the pseudo-labeled target data.
- **AT+UDA**. We train a UDA model where the source examples are all adversarial and the target examples are clean.
- **ARTUDA** (Lo & Patel, 2022). ARTUDA can be seen as a combination of DANN (Ganin & Lempitsky, 2015) and TRADES (Zhang et al., 2019a). In comparison to DART with clean source, ARTUDA applies two domain divergences to measure the discrepancy between clean source and clean target, as well as between clean source and adversarial target. Additionally, ARTUDA's methodology for generating adversarial target examples does not take the domain divergence into consideration, which differs from DART.
- **SRoUDA** (Zhu et al., 2023). SRoUDA alternates between adversarial training on target data with pseudo-labels and fine-tuning the pseudo-label predictor. The pseudo-label predictor has a similar role to that in DART; it is initially trained using a standard UDA method and is then continuously fine-tuned via a meta-step, a technique originally proposed by (Pham et al., 2021). Moreover, Zhu et al. (2023) introduced novel data augmentation methods such as random masked augmentation to further enhance robustness.
- **DART.** We experiment with DART for three different source choices as described in Section 4; namely, **DART(clean src)**, **DART(adv src)**, and **DART(kl src)**.

For fairness, we apply the same data augmentation scheme that is used in (Gulrajani & Lopez-Paz, 2020) (described in Appendix C.4) across all algorithms including SRoUDA.

**Architecture and optimization.** For DIGIT datasets, we consider multi-layer convolutional networks (see Table 5 in the appendix for the architecture). For the other datasets, we consider ResNet50 (pre-trained using ImageNet) as the backbone feature extractor and all batch normalization layers frozen. We consider a linear layer as the classifier on top of the feature extractor. We use cross-entropy loss and Adam (Kinga et al., 2015) for optimization. We first pre-train each model using DANN, while periodically evaluating the standard accuracy of different checkpoints during pre-training. We then pick the checkpoint with the highest standard accuracy and use it as an initialization for all algorithms. We use the same number of training iterations for pre-training and running the algorithms.

**Robustness setup.** We assume an $\ell_\infty$-norm perturbation set $\mathcal{B}(\mathrm{x}) = \{\tilde{\mathrm{x}} : \|\tilde{\mathrm{x}} - \mathrm{x}\|_\infty \leq \alpha\}$ and set $\alpha = 2/255$. During training, adversarial examples are generated using 5 steps of PGD with step size $1/255$. We evaluate all algorithms on the target data using standard accuracy and robust accuracy, computed using two different attack methods: (i) PGD attack with 20 iterations, and (ii) AutoAttack (Croce & Hein, 2020b), which includes four diverse attacks, namely APGD-CE, APGD-target, FAB (Croce & Hein, 2020a), and Square Attack (Andriushchenko et al., 2020). Note that these attack methods have full access to the model parameters (i.e., white-box attacks), and are constrained by the same perturbation size $\alpha$. If not specifically stated, we evaluate on $\alpha = 2/255$.

**Hyperparameter tuning.** We follow an oracle setting where a small labeled validation set from the target domain is used for tuning. This approach is commonly used for hyperparameter tuning in the literature on UDA (Long et al., 2013; Shen et al., 2018; Kumar et al., 2018; Wei & Hsu, 2018). If no labeled validation set is available, the oracle setting can be viewed as an upper bound on the performance of UDA methods. For the source domain, we keep 80% of the data (90% for VisDA). For the target domain, we split the data into unlabeled training data, validation data and test data with a ratio of 6:2:2 (8:1:1 for VisDA[4]). For each algorithm, we perform 20 random search trials[4] over the hyperparameter distribution (see Appendix C.5). We apply early stopping and select the best model amongst the 20 models from random search, based on its performance on the target validation set. We repeat the entire suite of experiments three times, reselecting random values

---

[4]As VisDA is a large dataset, we choose a different proportion and only perform 10 random search trials to save computational resources.

for hyperparameters, re-initializing the weights and redoing dataset splits each time. The reported results are the means over these three repetitions, along with their standard errors. This experimental setup resulted in training a total of 29700 models.

## 5.2 RESULTS

**Performance on benchmarks.** For each of the 4 benchmark datasets, we train and evaluate all algorithms on all possible source-target pairs. In Table 1, we report the results for each dataset, averaged over all corresponding source-target pairs (values after the $\pm$ sign are the standard error of the mean). We refer the reader to Appendix D for full results for each of the 46 source-target pairs.

| Dataset | | DIGIT (20 source-target pairs) | | | OfficeHome (12 source-target pairs) | | | PACS (12 source-target pairs) | | | VisDA (2 source-target pairs) | | |
|---|---|---|---|---|---|---|---|---|---|---|---|---|---|
| Algorithm | | nat acc | pgd acc | aa acc | nat acc | pgd acc | aa acc | nat acc | pgd acc | aa acc | nat acc | pgd acc | aa acc |
| No defense | Natural DANN | 69.9±0.3 | 53.9±0.5 | 53.4±0.5 | **57.4±0.2** | 1.5±0.1 | 0.4±0.0 | 81.1±0.3 | 11.0±0.2 | 3.6±0.2 | 73.0±0.4 | 0.6±0.1 | 0.0±0.0 |
| Source data only | AT(src only) | 71.5±0.1 | 62.0±0.1 | 61.7±0.1 | 49.7±0.7 | 31.2±0.1 | 29.9±0.2 | 65.7±0.9 | 48.2±0.1 | 47.0±0.1 | 36.2±0.5 | 29.8±0.3 | 28.5±0.3 |
| | TRADES(src only) | 71.1±0.0 | 62.4±0.0 | 62.0±0.0 | 48.4±0.4 | 31.5±0.2 | 30.1±0.2 | 66.1±0.7 | 48.2±0.3 | 45.9±0.3 | 36.5±0.2 | 29.5±0.6 | 28.9±0.6 |
| Target data + pseudo-label | AT(tgt,pseudo) | 73.8±0.1 | 68.9±0.1 | 68.6±0.0 | 52.7±0.1 | 40.5±0.2 | 39.8±0.2 | 82.0±0.4 | 70.0±0.2 | 69.6±0.3 | 77.7±0.2 | 70.2±0.3 | 69.6±0.3 |
| | TRADES(tgt,pseudo) | 73.9±0.1 | 69.8±0.0 | 69.4±0.0 | 53.0±0.5 | 41.4±0.3 | 40.6±0.3 | 82.7±0.2 | 71.7±0.3 | 71.1±0.4 | 76.6±0.4 | 69.7±0.1 | 69.1±0.1 |
| Robust UDA methods | AT+UDA | 71.9±0.1 | 63.0±0.1 | 62.7±0.1 | 51.3±0.9 | 32.7±0.1 | 31.2±0.2 | 68.6±0.8 | 52.9±0.9 | 44.4±0.1 | 57.2±0.7 | 36.7±0.3 | 33.2±0.7 |
| | ARTUDA | 74.3±0.2 | 70.6±0.1 | 70.3±0.1 | 54.6±0.3 | 39.0±0.5 | 37.1±0.6 | 74.6±0.3 | 60.5±0.2 | 58.1±0.6 | 58.9±1.3 | 47.6±1.3 | 46.2±1.5 |
| | SROUDA | 73.7±0.1 | 69.2±0.1 | 68.8±0.1 | 51.3±0.2 | 40.6±0.1 | 38.7±0.2 | 76.1±0.7 | 65.3±0.3 | 64.0±0.5 | 64.7±1.9 | 53.2±1.0 | 51.2±1.1 |
| DART | DART(clean src) | 78.3±0.2 | **74.5±0.1** | **74.4±0.1** | 56.4±0.1 | 40.7±0.1 | 39.6±0.1 | **85.5±0.1** | **73.3±0.0** | **72.6±0.1** | **78.4±0.1** | **71.7±0.2** | 71.3±0.3 |
| | DART(adv src) | 77.8±0.2 | 74.0±0.2 | 73.9±0.2 | 55.6±0.2 | **42.6±0.3** | **41.6±0.2** | 84.4±0.3 | 72.7±0.0 | 72.2±0.0 | 77.6±0.4 | 70.9±0.6 | 70.6±0.7 |
| | DART(kl src) | **78.3±0.1** | **74.5±0.1** | **74.4±0.1** | 56.0±0.2 | 42.4±0.2 | 41.3±0.2 | 85.3±0.2 | 73.1±0.3 | **72.6±0.4** | 78.2±0.5 | 71.3±0.7 | **71.9±0.4** |

Table 1: Standard accuracy (nat acc)/ Robust accuracy under PGD attack (pgd acc)/ Robust accuracy under AutoAttack (aa acc) on the target test data, averaged over all possible source-target pairs.

Based on Table 1, our method DART demonstrates significant improvements in adversarial robustness when compared to the various baselines. As expected, Natural DANN (which does not use any defense mechanism) has the lowest robust accuracy. Baselines that solely rely on the source data (specifically, AT(src only) and TRADES(src only)) display lower robustness compared to the other baselines, indicating that robustness does not transfer well due to the distribution shift.

Table 1 shows that DART consistently outperforms the robust UDA methods (AT+UDA, ARTUDA, and SROUDA), in terms of robust accuracy across all four benchmarks. It is essential to highlight that previous work investigating adversarial robustness in the UDA setting has not assessed two natural baselines we consider: AT(tgt,pseudo) and TRADES(tgt,pseudo). The latter two baselines appear to be very competitive with the robust UDA methods from the literature – but DART clearly outperforms these baselines. A more granular inspection of the results across the 46 source-target pairs (in Appendix D) reveals that DART consistently ranks first in terms of robust target test accuracy for 33 pairs under PGD attack and 35 pairs[5] under AutoAttack. The results also demonstrate that DART does not compromise standard accuracy. In fact, DART even improves standard accuracy on the DIGIT and PACS datasets, as indicated in Table 1. Across the entirety of the 46 source-target pairs, DART achieves the highest standard accuracy on 30 pairs.

**Ablation study.** We examine the effectiveness of the individual components in DART's objective function (Equation 7), by performing an ablation study on three source-target pairs: SVHN→MNIST, SYN→MNIST-M, and PACS for Photo→Sketch. Specifically, we consider DART(clean src) and study the following ablation scenarios: (1) w/o domain divergence term: we remove the second term $\Omega$ in the objective function; (2) w/o the approximation of the ideal joint worst target loss: we exclude the third term in the objective function; (3) we obtain pseudo-labels by standard UDA method, and fix it throughout the training process; (4) we use the current model to predict pseudo-labels for the third term. The results are presented in Table 2. Our findings reveal that omitting either the domain divergence or third term (which approximates the joint worst target loss) results in a significant performance degradation, confirming that these components are important. On the other hand, for DART without the pseudo-labeling technique discussed in Section 4, scenarios (3) and (4) still experienced some performance degradation in both standard and robust accuracy compared to DART. In summary, DART with all the components achieves the best performance.

**Comparison with additional baselines.** To strengthen the comparison, we propose two new, modified baselines that run adversarial training or TRADES on pseudo-labeled target data, where the pseudo labels are generated using exactly the pseudo labeling method used in our approach (described in Section 4)–we refer to these methods as AT(tgt,cg) and TRADES(tgt,cg). These two methods are similar to AT(tgt,pseudo) and TRADES(tgt,pseudo), with the main difference that the pseudo labels may change during training. We evaluate DART against AT(tgt,cg) and TRADES(tgt,cg) on the same source-target pairs: SVHN→MNIST, SYN→MNIST-M, and PACS for Photo→Sketch.

---

[5]In cases of a tie, we prioritize the algorithm with the smaller standard error.

| Source→Target | SVHN→MNIST | | | SYN→MNIST-M | | | PACS Photo→Sketch | | |
|---|---|---|---|---|---|---|---|---|---|
| Algorithm | nat acc | pgd acc | aa acc | nat acc | pgd acc | aa acc | nat acc | pgd acc | aa acc |
| (1) DART w/o DANN term | 95.0±0.1 | 90.9±0.4 | 90.8±0.4 | 67.2±0.7 | 45.1±0.6 | 44.8±0.6 | 79.1±0.7 | 76.3±0.7 | 76.1±0.7 |
| (2) DART w/o third term | 84.8±0.3 | 81.3±0.2 | 81.2±0.3 | 65.7±0.5 | 53.6±0.3 | 53.3±0.4 | 72.3±2.4 | 63.8±1.1 | 60.6±0.8 |
| (3) DART fixed label | 87.3±0.2 | 85.2±0.2 | 85.1±0.2 | 65.6±0.4 | 56.0±0.3 | 55.5±0.5 | 79.3±0.4 | 75.7±0.4 | 75.4±0.4 |
| (4) DART self label | 96.9±1.2 | 95.9±1.6 | 95.9±1.6 | 70.6±3.3 | 63.5±3.0 | 63.4±3.0 | 75.5±1.3 | 68.6±0.7 | 67.9±0.7 |
| DART w. all components | **98.7±0.1** | **98.2±0.2** | **98.2±0.2** | **75.2±0.8** | **66.7±0.8** | **66.5±0.8** | **82.5±0.8** | **79.9±0.4** | **79.5±0.5** |

Table 2: Standard accuracy (nat acc)/ Robust accuracy under PGD attack (pgd acc)/ Robust accuracy under AutoAttack (aa acc) on target test data for three source-target pairs.

The results, presented in Table 3, show that DART outperforms these baselines, suggesting that DART's good performance is not solely due to the proposed pseudo-labeling method.

| Source→Target | SVHN→MNIST | | | SYN→MNIST-M | | | PACS Photo→Sketch | | |
|---|---|---|---|---|---|---|---|---|---|
| Algorithm | nat acc | pgd acc | aa acc | nat acc | pgd acc | aa acc | nat acc | pgd acc | aa acc |
| AT(tgt,cg) | 91.4±0.1 | 90.2±0.1 | 90.2±0.1 | 67.7±0.7 | 58.6±0.6 | 58.4±0.6 | 79.6±0.5 | 76.3±0.7 | 75.9±0.7 |
| TRADES(tgt,cg) | 97.1±0.4 | 96.6±0.4 | 96.6±0.4 | 68.5±0.7 | 63.2±0.9 | 63.1±0.8 | 78.5±0.7 | 76.4±0.6 | 76.1±0.5 |
| DART (clean src) | **98.7±0.1** | 98.2±0.2 | 98.2±0.2 | **75.2±0.8** | **66.7±0.8** | **66.5±0.8** | **82.5±0.8** | **79.9±0.4** | **79.5±0.5** |
| DART (adv src) | **98.7±0.1** | **98.3±0.2** | **98.3±0.2** | 72.6±0.9 | 64.3±1.0 | 64.1±1.0 | 81.0±1.0 | 78.1±0.4 | 77.7±0.4 |
| DART (kl src) | 98.6±0.2 | 98.2±0.2 | 98.2±0.2 | 74.4±1.2 | 66.0±1.3 | 65.8±1.3 | 82.4±1.6 | 79.2±1.2 | 78.8±1.1 |

Table 3: Standard accuracy (nat acc) / Robust accuracy under PGD attack (pgd acc) / Robust accuracy under AutoAttack (aa acc) on target test data for three source-target pairs.

**Different domain divergence metrics.** As discussed earlier, DART is compatible with a wide class of UDA divergence metrics previously proposed in the literature. Here we study DART's performance on PACS Photo→Sketch when using alternative domain divergence metrics: MMD (Maximum Mean Discrepancy) (Gretton et al., 2012) and CORAL (Correlation Alignment) (Sun & Saenko, 2016). We conduct a similar investigation with AT+UDA, ARTUDA, and SROUDA. The results, reported in Table 4, demonstrate that DART consistently outperforms the adversarially robust UDA baselines for all three domain divergence metrics considered.

| Domain Divergence | DANN | | | MMD | | | CORAL | | |
|---|---|---|---|---|---|---|---|---|---|
| Algorithm | nat acc | pgd acc | aa acc | nat acc | pgd acc | aa acc | nat acc | pgd acc | aa acc |
| Natural UDA | 74.0±1.1 | 24.0±2.1 | 5.6±1.3 | 68.7±0.9 | 23.4±1.5 | 11.0±1.4 | 68.2±1.1 | 3.6±1.2 | 0.2±0.1 |
| AT+UDA | 70.6±1.6 | 62.2±1.5 | 60.9±1.5 | 73.3±0.3 | 66.3±0.3 | 65.7±0.3 | 67.0±3.5 | 55.7±1.7 | 53.8±2.4 |
| ARTUDA | 74.9±1.3 | 70.4±1.4 | 69.2±1.3 | 60.2±2.3 | 54.6±1.7 | 52.8±1.9 | 42.8±1.2 | 34.2±1.6 | 31.6±1.8 |
| SROUDA | 71.9±0.9 | 63.7±1.2 | 60.8±2.0 | 62.3±4.3 | 56.7±4.1 | 54.9±4.1 | 61.3±2.0 | 53.6±1.5 | 51.8±1.9 |
| DART(clean src) | 82.5±0.8 | 79.9±0.4 | 79.5±0.5 | 76.8±1.1 | 76.8±1.1 | 73.7±1.4 | 80.2±0.8 | 76.8±0.8 | 76.5±0.8 |
| DART(adv src) | 81.0±1.0 | 78.1±0.4 | 77.7±0.4 | 79.2±1.4 | 76.6±1.4 | 76.6±1.4 | 83.0±0.8 | 80.5±0.9 | 80.3±0.8 |
| DART(kl src) | 82.4±1.6 | 79.2±1.2 | 78.8±1.1 | 77.6±1.2 | 75.1±0.9 | 74.9±0.8 | 81.7±1.4 | 79.3±1.5 | 79.3±1.5 |

Table 4: Standard accuracy (nat acc) / Robust accuracy under PGD attack (pgd acc) / Robust accuracy under AutoAttack (aa acc) on target test data, for three different domain divergence metrics.

**Sanity checks on the PGD attack.** We present some sanity checks to demonstrate that no gradient masking phenomena (Athalye et al., 2018) exist in our setup. First, we increase the PGD attack strength (perturbation size) when evaluating the algorithms for PACS (Photo→Sketch); see Figure 1. It is evident from the figure that as the perturbation size increases, the robust accuracy of all the algorithms decreases, while DART consistently outperforms the other algorithms for all perturbation sizes. With sufficiently large perturbation size, the robust accuracy of all algorithms drops to zero, as expected. Second, we fix the perturbation size to 2/255, and gradually increase the number of attack iterations–we present the results of this experiment in Figure 2 in the appendix. The results of Figure 2 indicate that 20 attack iterations are sufficient to achieve the strongest PGD attack within the given perturbation size.

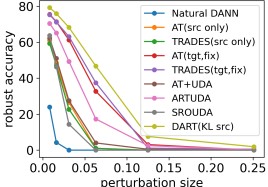

Figure 1: Robust accuracy as a function of perturbation size for different algorithms on PACS (Photo→Sketch).

# 6 CONCLUSION

In this paper, we tackled the problem of learning adversarially robust models under an unsupervised domain adaptation setting. We developed robust generalization guarantees and provided a unified, practical defense framework (DART), which can be integrated with standard UDA methods. We also released a new testbed (DomainRobust) and performed extensive experiments, which demonstrate that DART consistently outperforms the state of the art across four multi-domain benchmarks. A natural next step is to extend our theory and defense framework to other settings of distribution shift such as domain generalization.

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

# Supplementary Material

## A PROOF OF THEOREM 3.1

Before presenting the proof, we first define *the set of all possible perturbed distributions* as follows:

$$\mathcal{P}(\mathcal{D}_T) = \left\{ \tilde{\mathcal{D}} : (T(\mathrm{x}), y) \sim \tilde{\mathcal{D}}, \exists \text{ a map } T : \mathcal{X} \to \mathcal{X} \text{ with } T(\mathrm{x}) \in \mathcal{B}(\mathrm{x}), (\mathrm{x}, y) \sim \mathcal{D}_T \right\}. [6]$$

As an illustrative example, consider the scenario where the perturbation set $\mathcal{B}$ is defined as an $\ell_p$ norm ball, then set of perturbation distributions $\mathcal{P}$ can be effectively constructed using the Wasserstein metric (Staib & Jegelka, 2017; Khim & Loh, 2018; Regniez et al., 2021).

**Theorem 3.1.** Let $\mathcal{H}$ be a hypothesis class with finite VC dimension $\mathrm{VC}(\mathcal{H})$ and adversarial VC dimension $\mathrm{AVC}(\mathcal{H})$ (Cullina et al., 2018). If $\mathcal{Z}_S$ and $\mathcal{Z}_T$ are labeled samples of size[7] $n$ drawn i.i.d. from $\mathcal{D}_S$ and $\mathcal{D}_T$, respectively, and $\mathrm{X}_S$ and $\mathrm{X}_T$ are the corresponding data matrices, then for any $\delta \in (0, 1)$, w.p. at least $1 - \delta$, for all $h \in \mathcal{H}$,

$$L_{\mathrm{adv}}^{0/1}(h; \mathcal{D}_T) \leq \underbrace{L^{0/1}(h; \mathcal{Z}_S)}_{\text{Source Loss}} + \underbrace{\sup_{\substack{\tilde{\mathrm{x}}_i^t \in \mathcal{B}(\mathrm{x}_i^t), \forall i \in [n], \\ \tilde{\mathcal{Z}}_T = \left\{ (\tilde{\mathrm{x}}_i^t, y_i^t) \right\}_{i=1}^n}}}_{\text{Worst-case target}} \left[ \underbrace{d_{\mathcal{H}\Delta\mathcal{H}}(\mathrm{X}_S, \tilde{\mathrm{X}}_T)}_{\text{Domain Divergence}} + 2 \underbrace{\gamma(\mathcal{Z}_S, \tilde{\mathcal{Z}}_T)}_{\text{Ideal Joint Loss}} \right] + \epsilon, \quad (2)$$

where the generalization gap $\epsilon = \mathcal{O}(\sqrt{\frac{\max\{\mathrm{VC}(\mathcal{H}), \mathrm{AVC}(\mathcal{H})\} \log(n) + \log(1/\delta)}{n}})$, the (empirical) ideal joint loss is defined as $\gamma(\mathcal{Z}_S, \mathcal{Z}_T) := \min_{h^* \in \mathcal{H}} \left[ L^{0/1}(h^*; \mathcal{Z}_S) + L^{0/1}(h^*; \mathcal{Z}_T) \right]$, and the (empirical) $\mathcal{H}\Delta\mathcal{H}$-divergence can be computed as follows[8]:

$$d_{\mathcal{H}\Delta\mathcal{H}}(\mathrm{X}_S, \mathrm{X}_T) = 2 \left( 1 - \min_{h \in \mathcal{H}\Delta\mathcal{H}} \left[ \frac{1}{n} \sum_{\mathrm{x}: h(\mathrm{x}) = 0} \mathbb{1}(\mathrm{x} \in \mathrm{X}_S) + \frac{1}{n} \sum_{\mathrm{x}: h(\mathrm{x}) = 1} \mathbb{1}(\mathrm{x} \in \mathrm{X}_T) \right] \right). \quad (3)$$

*Proof of Theorem 3.1.* We use the notation $f_S$ and $f_T$ to represent labeling function $\mathcal{X} \to \mathcal{Y}$ of the given source domain and target domain. Note that $(\mathrm{x}, y) \sim \mathcal{D}_S$ implies $f_S(\mathrm{x}) = y$, and $(\mathrm{x}, y) \sim \mathcal{D}_T$ implies $f_T(\mathrm{x}) = y$. Here we consider 0-1 loss, which can be represented as $\ell(y_1, y_2) = |y_1 - y_2|$.

For any given $\mathcal{D}_T$, we consider $h^* := h^*(\mathcal{D}_T) = \mathrm{argmin}_{h \in \mathcal{H}} \gamma(\mathcal{D}_S, \mathcal{D}_T)$. We first upper bound the adversarial target risk as follows:

$$
\begin{aligned}
&L_{\mathrm{adv}}^{0/1}(h; \mathcal{D}_T) \\
&= \mathbb{E}_{(\mathrm{x}, y) \sim \mathcal{D}_T} \sup_{\tilde{\mathrm{x}} \in \mathcal{B}(\mathrm{x})} |h(\tilde{\mathrm{x}}) - f_T(\mathrm{x})| \\
&= \sup_{\tilde{\mathcal{D}}_T \in \mathcal{P}(\mathcal{D}_T)} \mathbb{E}_{(\mathrm{x}, y) \sim \tilde{\mathcal{D}}_T} |h(\mathrm{x}) - f_T(\mathrm{x})| \qquad\qquad\qquad \text{(By the definition of } \mathcal{P}(\mathcal{D}_T).) \\
&\leq \sup_{\tilde{\mathcal{D}}_T \in \mathcal{P}(\mathcal{D}_T)} \mathbb{E}_{(\mathrm{x}, y) \sim \tilde{\mathcal{D}}_T} |h^*(\mathrm{x}) - f_T(\mathrm{x})| + \sup_{\tilde{\mathcal{D}}_T \in \mathcal{P}(\mathcal{D}_T)} \mathbb{E}_{(\mathrm{x}, y) \sim \tilde{\mathcal{D}}_T} |h(\mathrm{x}) - h^*(\mathrm{x})| \\
&= \sup_{\tilde{\mathcal{D}}_T \in \mathcal{P}(\mathcal{D}_T)} \mathbb{E}_{(\mathrm{x}, y) \sim \tilde{\mathcal{D}}_T} |h^*(\mathrm{x}) - f_T(\mathrm{x})| + \mathbb{E}_{(\mathrm{x}, y) \sim \mathcal{D}_S} |h(\mathrm{x}) - h^*(\mathrm{x})| \\
&\qquad\qquad + \sup_{\tilde{\mathcal{D}}_T \in \mathcal{P}(\mathcal{D}_T)} \mathbb{E}_{(\mathrm{x}, y) \sim \tilde{\mathcal{D}}_T} |h(\mathrm{x}) - h^*(\mathrm{x})| - \mathbb{E}_{(\mathrm{x}, y) \sim \mathcal{D}_S} |h(\mathrm{x}) - h^*(\mathrm{x})| \\
&\leq \sup_{\tilde{\mathcal{D}}_T \in \mathcal{P}(\mathcal{D}_T)} \mathbb{E}_{(\mathrm{x}, y) \sim \tilde{\mathcal{D}}_T} |h^*(\mathrm{x}) - f_T(\mathrm{x})| + \mathbb{E}_{(\mathrm{x}, y) \sim \mathcal{D}_S} |h(\mathrm{x}) - h^*(\mathrm{x})|
\end{aligned}
$$

---

[6] We slightly abuse the notation when the input distribution is over $\mathcal{X}$; i.e., $\mathcal{P}(\mathcal{D}_T^X) = \left\{ \tilde{\mathcal{D}} : T(\mathrm{x}) \sim \tilde{\mathcal{D}}, \exists \text{ a map } T : \mathcal{X} \to \mathcal{X} \text{ with } T(\mathrm{x}) \in \mathcal{B}(\mathrm{x}), \mathrm{x} \sim \mathcal{D}_T^X \right\}$.

[7] We assume that $\mathcal{Z}_S$ and $\mathcal{Z}_T$ have the same size for simplicity. The result still applies to different sizes.

[8] In Definition 2.1, we defined $d_{\mathcal{H}\Delta\mathcal{H}}$ for two input distributions. Here we use an equivalent definition in which the two inputs are data matrices.

$$+ \left| \sup_{\tilde{\mathcal{D}}_T \in \mathcal{P}(\mathcal{D}_T)} \mathbb{E}_{(\mathrm{x},y) \sim \tilde{\mathcal{D}}_T} |h(\mathrm{x}) - h^*(\mathrm{x})| - \mathbb{E}_{(\mathrm{x},y) \sim \mathcal{D}_S} |h(\mathrm{x}) - h^*(\mathrm{x})| \right|$$

$$\leq \sup_{\tilde{\mathcal{D}}_T \in \mathcal{P}(\mathcal{D}_T)} \mathbb{E}_{(\mathrm{x},y) \sim \tilde{\mathcal{D}}_T} |h^*(\mathrm{x}) - f_T(\mathrm{x})| + \mathbb{E}_{(\mathrm{x},y) \sim \mathcal{D}_S} |h^*(\mathrm{x}) - f_S(\mathrm{x})| + \mathbb{E}_{(\mathrm{x},y) \sim \mathcal{D}_S} |h(\mathrm{x}) - f_S(\mathrm{x})|$$

$$+ \left| \sup_{\tilde{\mathcal{D}}_T \in \mathcal{P}(\mathcal{D}_T)} \mathbb{E}_{(\mathrm{x},y) \sim \tilde{\mathcal{D}}_T} |h(\mathrm{x}) - h^*(\mathrm{x})| - \mathbb{E}_{(\mathrm{x},y) \sim \mathcal{D}_S} |h(\mathrm{x}) - h^*(\mathrm{x})| \right|$$

$$= \sup_{\tilde{\mathcal{D}}_T \in \mathcal{P}(\mathcal{D}_T)} L^{0/1}(h^*; \tilde{\mathcal{D}}_T) + L^{0/1}(h^*; \mathcal{D}_S) + L^{0/1}(h; \mathcal{D}_S)$$

$$+ \left| \sup_{\tilde{\mathcal{D}}_T \in \mathcal{P}(\mathcal{D}_T)} \mathbb{E}_{(\mathrm{x},y) \sim \tilde{\mathcal{D}}_T} |h(\mathrm{x}) - h^*(\mathrm{x})| - \mathbb{E}_{(\mathrm{x},y) \sim \mathcal{D}_S} |h(\mathrm{x}) - h^*(\mathrm{x})| \right|$$

$$= L^{0/1}(h; \mathcal{D}_S) + \sup_{\tilde{\mathcal{D}}_T \in \mathcal{P}(\mathcal{D}_T)} \gamma(\mathcal{D}_S, \tilde{\mathcal{D}}_T)$$

$$+ \left| \sup_{\tilde{\mathcal{D}}_T \in \mathcal{P}(\mathcal{D}_T)} \mathbb{E}_{(\mathrm{x},y) \sim \tilde{\mathcal{D}}_T} |h(\mathrm{x}) - h^*(\mathrm{x})| - \mathbb{E}_{(\mathrm{x},y) \sim \mathcal{D}_S} |h(\mathrm{x}) - h^*(\mathrm{x})| \right|$$

$$\leq L^{0/1}(h; \mathcal{D}_S) + \sup_{\tilde{\mathcal{D}}_T \in \mathcal{P}(\mathcal{D}_T)} \gamma(\mathcal{D}_S, \tilde{\mathcal{D}}_T)$$

$$+ \sup_{h \in \mathcal{H} \Delta \mathcal{H}} \left| \sup_{\tilde{\mathcal{D}}_T \in \mathcal{P}(\mathcal{D}_T)} \mathbb{E}_{(\mathrm{x},y) \sim \tilde{\mathcal{D}}_T} \mathbb{1}\left[ h(\mathrm{x}) = 1 \right] - \mathbb{E}_{(\mathrm{x},y) \sim \mathcal{D}_S} \mathbb{1}\left[ h(\mathrm{x}) = 1 \right] \right|$$

(Denote this line as $\frac{1}{2} d_{\mathcal{H} \Delta \mathcal{H}_{\mathrm{adv}}}(\mathcal{D}_S^X, \mathcal{D}_T^X)$, which we also define later in the proof.)

$$= L^{0/1}(h; \mathcal{D}_S) + \sup_{\tilde{\mathcal{D}}_T \in \mathcal{P}(\mathcal{D}_T)} \gamma(\mathcal{D}_S, \tilde{\mathcal{D}}_T) + \frac{1}{2} d_{\mathcal{H} \Delta \mathcal{H}_{\mathrm{adv}}}(\mathcal{D}_S^X, \mathcal{D}_T^X) \tag{8}$$

Given distributions $\mathcal{D}_S, \mathcal{D}_T$, samples $\mathrm{X}_S, \mathrm{X}_T$ each with size $n$, we recall the definition of expected and empirical $\mathcal{H} \Delta \mathcal{H}$-divergence:

$$d_{\mathcal{H} \Delta \mathcal{H}}(\mathcal{D}_S^X, \mathcal{D}_T^X) = 2 \sup_{h \in \mathcal{H} \Delta \mathcal{H}} \left| \mathbb{E}_{\mathrm{x} \sim \mathcal{D}_S^X} \mathbb{1}\left[ h(\mathrm{x}) = 1 \right] - \mathbb{E}_{\mathrm{x} \sim \mathcal{D}_T^X} \mathbb{1}\left[ h(\mathrm{x}) = 1 \right] \right|$$

$$d_{\mathcal{H} \Delta \mathcal{H}}(\mathrm{X}_S, \mathrm{X}_T) = 2 \sup_{h \in \mathcal{H} \Delta \mathcal{H}} \left| \frac{1}{n} \sum_{i=1}^{n} \mathbb{1}\left[ h(\mathrm{x}_i^s) = 1 \right] - \frac{1}{n} \sum_{i=1}^{n} \mathbb{1}\left[ h(\mathrm{x}_i^t) = 1 \right] \right|$$

We now define the expected and empirical adversarial target domain divergence, namely $d_{\mathcal{H} \Delta \mathcal{H}_{\mathrm{adv}}}(\mathcal{D}_S^X, \mathcal{D}_T^X)$ and $\hat{d}_{\mathcal{H} \Delta \mathcal{H}_{\mathrm{adv}}}(\mathrm{X}_S, \mathrm{X}_T)$, respectively, as follows:

$$d_{\mathcal{H} \Delta \mathcal{H}_{\mathrm{adv}}}(\mathcal{D}_S^X, \mathcal{D}_T^X) = 2 \sup_{h \in \mathcal{H} \Delta \mathcal{H}} \left| \sup_{\tilde{\mathcal{D}}_T \in \mathcal{P}(\mathcal{D}_T)} \mathbb{E}_{(\mathrm{x},y) \sim \tilde{\mathcal{D}}_T} \mathbb{1}\left[ h(\mathrm{x}) = 1 \right] - \mathbb{E}_{(\mathrm{x},y) \sim \mathcal{D}_S} \mathbb{1}\left[ h(\mathrm{x}) = 1 \right] \right|$$

$$d_{\mathcal{H} \Delta \mathcal{H}_{\mathrm{adv}}}(\mathrm{X}_S, \mathrm{X}_T) = 2 \sup_{h \in \mathcal{H} \Delta \mathcal{H}} \left| \frac{1}{n} \sum_{i=1}^{n} \sup_{\tilde{\mathrm{x}}_i^t \in \mathcal{B}(\mathrm{x}_i^t)} \mathbb{1}\left[ h(\tilde{\mathrm{x}}_i^t) = 1 \right] - \frac{1}{n} \sum_{i=1}^{n} \mathbb{1}\left[ h(\mathrm{x}_i^s) = 1 \right] \right|$$

Under the same perturbation constraints, we have

$$d_{\mathcal{H} \Delta \mathcal{H}_{\mathrm{adv}}}(\mathcal{D}_S^X, \mathcal{D}_T^X) \leq \sup_{\tilde{\mathcal{D}}_T \in \mathcal{P}(\mathcal{D}_T)} d_{\mathcal{H} \Delta \mathcal{H}}(\mathcal{D}_S^X, \tilde{\mathcal{D}}_T^X)$$

$$d_{\mathcal{H} \Delta \mathcal{H}_{\mathrm{adv}}}(\mathrm{X}_S, \mathrm{X}_T) \leq \sup_{\tilde{\mathrm{X}}_T \in \mathcal{P}(\mathrm{X}_T)} d_{\mathcal{H} \Delta \mathcal{H}}(\mathrm{X}_S, \tilde{\mathrm{X}}_T) \tag{9}$$

Therefore, the following upper bound holds:

$$d_{\mathcal{H} \Delta \mathcal{H}_{\mathrm{adv}}}(\mathcal{D}_S^X, \mathcal{D}_T^X) - d_{\mathcal{H} \Delta \mathcal{H}_{\mathrm{adv}}}(\mathrm{X}_S, \mathrm{X}_T)$$

$$\leq 2 \sup_{h \in \mathcal{H} \Delta \mathcal{H}} \left| \sup_{\tilde{\mathcal{D}}_T \in \mathcal{P}(\mathcal{D}_T)} \mathbb{E}_{(\mathrm{x},y) \sim \tilde{\mathcal{D}}_T} \mathbb{1}\left[ h(\mathrm{x}) = 1 \right] - \mathbb{E}_{(\mathrm{x},y) \sim \mathcal{D}_S} \mathbb{1}\left[ h(\mathrm{x}) = 1 \right] \right.$$

$$- \frac{1}{n} \sum_{i=1}^{n} \sup_{\tilde{x}_i^t \in \mathcal{B}(x_i^t)} \mathbb{1}\left[h(\tilde{x}_i^t) = 1\right] + \frac{1}{n} \sum_{i=1}^{n} \mathbb{1}\left[h(x_i^s) = 1\right] \Bigg|$$

$$\leq 2 \sup_{h \in \mathcal{H}\Delta\mathcal{H}} \Bigg| \sup_{\tilde{\mathcal{D}}_T \in \mathcal{P}(\mathcal{D}_T)} \mathbb{E}_{(x,y)\sim\tilde{\mathcal{D}}_T} \mathbb{1}\left[h(x) = 1\right] - \frac{1}{n} \sum_{i=1}^{n} \sup_{\tilde{x}_i^t \in \mathcal{B}(x_i^t)} \mathbb{1}\left[h(\tilde{x}_i^t) = 1\right] \Bigg|$$

$$\text{(Denote this line as } d_{\mathcal{H}_{\text{adv}}\Delta\mathcal{H}_{\text{adv}}}(\tilde{\mathcal{D}}_T^X, \tilde{X}_T))$$

$$+ 2 \sup_{h \in \mathcal{H}\Delta\mathcal{H}} \Bigg| \mathbb{E}_{(x,y)\sim\mathcal{D}_S} \mathbb{1}\left[h(x) = 1\right] - \frac{1}{n} \sum_{i=1}^{n} \mathbb{1}\left[h(x_i^s) = 1\right] \Bigg|$$

$$= d_{\mathcal{H}_{\text{adv}}\Delta\mathcal{H}_{\text{adv}}}(\tilde{\mathcal{D}}_T^X, \tilde{X}_T) + d_{\mathcal{H}\Delta\mathcal{H}}(\mathcal{D}_S^X, X_S)$$

Therefore from standard VC theory (Vapnik, 1999) we have

$$\mathbf{P}\left[ \sup_{h \in \mathcal{H}} \left| L^{0/1}(h; \mathcal{D}_S) - L^{0/1}(h; \mathcal{Z}_S) \right| \geq \frac{\epsilon}{4} \right] \leq 8(2n)^{\text{VC}(\mathcal{H})} \exp(-\frac{n\epsilon^2}{128}) \tag{10}$$

$$\mathbf{P}\left[ d_{\mathcal{H}\Delta\mathcal{H}}(\mathcal{D}_S^X, X_S) \geq \frac{\epsilon}{4} \right] \leq 8(2n)^{\text{VC}(\mathcal{H})} \exp(-\frac{n\epsilon^2}{128}) \tag{11}$$

Based on the adversarial VC theory from (Cullina et al., 2018), we have

$$\mathbf{P}\left[ d_{\mathcal{H}_{\text{adv}}\Delta\mathcal{H}_{\text{adv}}}(\tilde{\mathcal{D}}_T^X, \tilde{X}_T) \geq \frac{\epsilon}{4} \right] \leq 8(2n)^{\text{AVC}(\mathcal{H})} \exp(-\frac{n\epsilon^2}{128}) \tag{12}$$

Similarly, we recall the definition of expected and empirical ideal joint risk as follows:

$$\gamma(\mathcal{D}_S, \mathcal{D}_T) = \min_{h \in \mathcal{H}} \left[ L^{0/1}(h; \mathcal{D}_S) + L^{0/1}(h; \mathcal{D}_T) \right]$$

$$\gamma(\mathcal{Z}_S, \mathcal{Z}_T) = \min_{h \in \mathcal{H}} \left[ L^{0/1}(h; \mathcal{Z}_S) + L^{0/1}(h; \mathcal{Z}_T) \right]$$

We then have,

$$\gamma(\mathcal{D}_S, \mathcal{D}_T) - \gamma(\mathcal{Z}_S, \mathcal{Z}_T)$$

$$= \min_{h_1 \in \mathcal{H}} \left[ L^{0/1}(h_1; \mathcal{D}_S) + L^{0/1}(h_1; \mathcal{D}_T) \right] - \min_{h_2 \in \mathcal{H}} \left[ L^{0/1}(h_2; \mathcal{Z}_S) + L^{0/1}(h_2; \mathcal{Z}_T) \right]$$

$$\leq \left[ L^{0/1}(h_2; \mathcal{D}_S) + L^{0/1}(h_2; \mathcal{D}_T) \right] - \min_{h_2 \in \mathcal{H}} \left[ L^{0/1}(h_2; \mathcal{Z}_S) + L^{0/1}(h_2; \mathcal{Z}_T) \right]$$

$$\leq \sup_{h \in \mathcal{H}} \left[ L^{0/1}(h; \mathcal{D}_S) + L^{0/1}(h; \mathcal{D}_T) - L^{0/1}(h; \mathcal{Z}_S) - L^{0/1}(h; \mathcal{Z}_T) \right]$$

$$\leq \sup_{h \in \mathcal{H}} \left[ L^{0/1}(h; \mathcal{D}_S) - L^{0/1}(h; \mathcal{Z}_S) \right] + \sup_{h \in \mathcal{H}} \left[ L^{0/1}(h; \mathcal{D}_T) - L^{0/1}(h; \mathcal{Z}_T) \right]$$

Applying standard VC theory and adversarial VC theory leads to:

$$\mathbf{P}\left[ |\gamma(\mathcal{D}_S, \mathcal{D}_T) - \gamma(\mathcal{Z}_S, \mathcal{Z}_T)| \geq \frac{\epsilon}{4} \right]$$

$$\leq \mathbf{P}\left[ \sup_{h \in \mathcal{H}} \left| L^{0/1}(h; \mathcal{D}_S) - L^{0/1}(h; \mathcal{Z}_S) \right| \geq \frac{\epsilon}{8} \right] \cdot \mathbf{P}\left[ \sup_{h \in \mathcal{H}} \left| L^{0/1}(h; \mathcal{D}_T) - L^{0/1}(h; \mathcal{Z}_T) \right| \geq \frac{\epsilon}{8} \right]$$

$$\leq 64(2n)^{2\text{VC}(\mathcal{H})} \exp(-\frac{n\epsilon^2}{256}) \tag{13}$$

Consider each of the above events (10), (11), (12), (13) hold with probability $\frac{\delta}{4}$, we set $\epsilon = \mathcal{O}\left( \sqrt{\frac{\max(\text{VC}(\mathcal{H}), \text{AVC}(\mathcal{H})) \log(n) + \log(1/\delta)}{n}} \right)$. By taking a union bound over the above events gives us that with probability at least $1 - \delta$, the following holds:

$$L_{\text{adv}}^{0/1}(h; \mathcal{D}_T)$$

$$\leq L^{0/1}(h; \mathcal{D}_S) + \sup_{\tilde{\mathcal{D}}_T \in \mathcal{P}(\mathcal{D}_T)} \gamma(\mathcal{D}_S, \mathcal{D}_T) + \frac{1}{2} d_{\mathcal{H}\Delta\mathcal{H}_{\text{adv}}}(\mathcal{D}_S^X, \mathcal{D}_T^X) \qquad \text{(Equation (8))}$$

$$\leq L^{0/1}(h; \mathcal{Z}_S) + \sup_{\tilde{\mathcal{D}}_T \in \mathcal{P}(\mathcal{D}_T)} \gamma(\mathcal{D}_S, \mathcal{D}_T) + \frac{1}{2} d_{\mathcal{H}\Delta\mathcal{H}_{\text{adv}}}(\mathrm{X}_S, \mathrm{X}_T) + \epsilon \qquad \text{(Union bound)}$$

$$\leq L^{0/1}(h; \mathcal{Z}_S) + \sup_{\tilde{\mathcal{D}}_T \in \mathcal{P}(\mathcal{D}_T)} \gamma(\mathcal{D}_S, \mathcal{D}_T) + \frac{1}{2} \sup_{\tilde{\mathrm{X}}_T \in \mathcal{P}(\mathrm{X}_T)} d_{\mathcal{H}\Delta\mathcal{H}}(\mathrm{X}_S, \tilde{\mathrm{X}}_T) + \epsilon \qquad \text{(Equation (9))}$$

$$\leq L^{0/1}(h; \mathcal{Z}_S) + \sup_{\tilde{\mathrm{X}}_T \in \mathcal{P}(\mathrm{X}_T)} \left[ 2\gamma(\mathcal{Z}_S, \mathcal{Z}_T) + d_{\mathcal{H}\Delta\mathcal{H}}(\mathrm{X}_S, \tilde{\mathrm{X}}_T) \right] + \epsilon$$

(Given non-negative functions $a(x)$ and $b(x)$, $\sup_x a(x) + \sup_x b(x) \leq 2\sup_x(a(x) + b(x))$)

We remark that the theorem can be extended to any symmetric loss function that satisfies the triangle inequality.

$\square$

# B  DART DISCUSSION

## B.1  USING DART TO ROBUSTIFY DANN AGAINST $\ell_p$ ATTACKS

Here we provide a concrete example of DART, using DANN as the base UDA method, and assuming the standard (white-box) $\ell_p$ threat model with perturbation set $\mathcal{B}(\mathrm{x}) = \{\tilde{\mathrm{x}} : \|\tilde{\mathrm{x}} - \mathrm{x}\|_p \leq \alpha\}$ for some positive scalars $p$ and $\alpha$. In DANN, let $g$ be the feature extractor, $f$ be the network's label predictor, and $d$ be the domain classifier (a.k.a. discriminator), which approximates the divergence between the two domains. With this notation, the empirical proxy for domain divergence $\Omega$ can be written as $\Omega_{\text{DANN}}(\mathrm{X}_S, \mathrm{X}_T, g, d) = -\frac{1}{n_s}\sum_{i=1}^{n_s} \ell((d \circ g)(\mathrm{x}_i^s), 1) - \frac{1}{n_t}\sum_{i=1}^{n_t} \ell((d \circ g)(\mathrm{x}_i^t), 0)$, which represents the negated loss of the domain classifier $d$ (which classifies source domain examples as 1 and target examples as 0). To find a robust DANN against $\ell_p$ attacks, Equation (7) can be written more explicitly as:

$$\min_{f,g} \sup_d \sup_{\|\tilde{\mathrm{x}}_i^t - \mathrm{x}_i^t\|_p \leq \alpha, \forall i} \frac{1}{n_s} \sum_{i=1}^{n_s} \ell((f \circ g)(\tilde{\mathrm{x}}_i^s), y_i^s) + \lambda_2 \frac{1}{n_t} \sum_{i=1}^{n_t} \ell((f \circ g)(\tilde{\mathrm{x}}_i^t), h_p(\mathrm{x}_i^t))$$
$$- \lambda_1 \left( \frac{1}{n_s} \sum_{i=1}^{n_s} \ell((d \circ g)(\tilde{\mathrm{x}}_i^s), 1) + \frac{1}{n_t} \sum_{i=1}^{n_t} \ell((d \circ g)(\tilde{\mathrm{x}}_i^t), 0) \right)$$

where $h_p$ is a pseudo-label predictor. $\tilde{\mathrm{x}}_i^s$ can be chosen as 1) clean source $\tilde{\mathrm{x}}_i^s = \mathrm{x}_i^s$, 2) adversarial source $\tilde{\mathrm{x}}_i^s = \text{argmax}_{\|\tilde{\mathrm{x}}_i - \mathrm{x}_i^s\|_p \leq \epsilon} \ell(h(\tilde{\mathrm{x}}_i); y_i^s)$, or 3) KL source $\tilde{\mathrm{x}}_i^s = \text{argmax}_{\|\tilde{\mathrm{x}}_i - \mathrm{x}_i^s\|_p \leq \epsilon} \text{KL}((f \circ g)(\tilde{\mathrm{x}}_i), (f \circ g)(\mathrm{x}_i^s))$.

One common strategy for solving the problem above is by alternating optimization where we iterate between: (i) optimizing for transformed source and target data $\tilde{\mathrm{x}}_i^s$ and $\tilde{\mathrm{x}}_i^t$ for all $i$, (ii) optimizing over the domain divergence $d$, (iii) optimizing the neural network $f$ and $g$. The optimization problem over the neural network's weights $(f, g, d)$ can be done using gradient based methods such as SGD. Optimization over the transformed data $\tilde{\mathrm{X}}_S$ and $\tilde{\mathrm{X}}_T$ can be done using a wide range of constrained optimization methods (Bertsekas, 2016), such as projected gradient descent (PGD).

## B.2 Pseudocode of DART

---

**Algorithm 1** **D**ivergence-**A**ware adve**R**sarial **T**raining (DART)

---

**Require:** Labeled source training data $\{(x_i^s, y_i^s)\}_{i=1}^{n_s}$, unlabeled target training data $X_T = \{x_i^t\}_{i=1}^{n_t}$.
Feature extractor $g$, target classifier $f$. Training iteration $T$. Checkpoint frequency $K$. Pseudo-labeling approach.

1: Pre-train $f, g$ using Equation (4).
2: Calculate pseudo-label $\hat{Y}_T$ for unlabeled target training data.
3: **for** $t = 1, 2, \ldots T$ **do**
4:     Sample a random mini-batch of source and target examples with the same batch size.
5:     Choose either clean source examples or apply one of the following two transformations to the source examples: adversarial or KL.
6:     Update $f, g$ by optimizing over Equation (7).
7:     **if** $t \% K = 0$ **then**
8:         If the pseudo labeling approach chosen can generate new pseudo labels during training, update the pseudo-labels $\hat{Y}_T$ for the unlabeled target training data. Otherwise, keep using the initial pseudo labels.
9:     **end if**
10: **end for**
11: Return $f \circ g$.

---

## B.3 Pseudo labeling in DART

DART can use any pseudo labeling approach from the literature. Here we present a simple approach that we used in the experiments. We assume that we are given a proxy that can be used to evaluate the model's accuracy (standard or robust)–this is the same proxy used for hyperparameter tuning. We maintain a pseudo-label predictor $h_p$ (with the same model architecture as $f \circ g$). In step 2 of Algorithm 1, we assign weights of $f \circ g$ to $h_p$, and generate pseudo-labels for the target data $\hat{Y}_T = h_p(X_T)$. In step 8 of Algorithm 1, we approximate the standard accuracy of $f \circ g$ (using the proxy). If the accuracy is better than that of the current pseudo-label predictor, we update the pseudo-label predictor's ($h_p$) weights to that of $f \circ g$; otherwise, the pseudo-label predictor's ($h_p$) weights remain unchanged. We then regenerate the pseudo-labels $\hat{Y}_T = h_p(X_T)$.

## C Experimental Details

### C.1 Datasets

- DIGIT contains 5 popular digit datasets. In our implementation, we use the digit-five dataset presented by (Peng et al., 2019)
  1. MNIST is a dataset of greyscale handwritten digits. We include 64015 images.
  2. MNIST-M is created by combining MNIST digits with the patches randomly extracted from color photos of BSDS500 as their background. We include 64015 images.
  3. SVHN contains RGB images of printed digits cropped from pictures of house number plates. We include 96322 images.
  4. Synthetic digits contains synthetically generated images of English digits embedded on random backgrounds. We include 33075 images.
  5. USPS is a grayscale dataset automatically scanned from envelopes by the U.S. Postal Service. We include 9078 images.

- OfficeHome contains objects commonly found in office and home environments. It consists of images from 4 different domains: Artistic (2,427 images), Clip Art (4,365 images), Product (4,439 images) and Real-World (4,357 images). For each domain, the dataset contains images of 65 object categories.

- PACS is created by intersecting the classes found in Caltech256 (Photo), Sketchy (Photo, Sketch), TU-Berlin (Sketch) and Google Images (Art painting, Cartoon, Photo). It consists of four do-

mains, namely Photo (1,670 images), Art Painting (2,048 images), Cartoon (2,344 images) and Sketch (3,929 images). Each domain contains seven categories.

- VisDA is a synthetic-to-real dataset consisting of two parts: synthetic and real. The synthetic dataset contains 152,397 images generated by 3D rendering. The real dataset is built from the Microsoft COCO training and validation splits, resulting in a collection of 55,388 object images that correspond to 12 classes.

## C.2 COMMON DOMAIN DIVERGENCE IN UDA METHODS

Recall the following notation: $f$ represents the classifier, $g$ represents the feature extractor, $d$ represents the discriminator.

1. **Domain Adversarial Neural Network (DANN) (Ganin & Lempitsky, 2015).**
$$\Omega(\mathbf{X}_S, \mathbf{X}_T, g, d) = \sup_d \left( \mathbb{E}_{\mathbf{x}^s \in \mathbf{X}_S} \log(d \circ g(\mathbf{x}^s)) + \mathbb{E}_{\mathbf{x}^t \in \mathbf{X}_T} \log(1 - d \circ g(\mathbf{x}^t)) \right).$$

2. **Maximum Mean Discrepancy (MMD) (Gretton et al., 2012).** Given kernel $k(\cdot, \cdot)$,
$$\Omega(\mathbf{X}_S, \mathbf{X}_T, g, k)$$
$$= \mathbb{E}_{\mathbf{x}^s \in \mathbf{X}_S} k(g(\mathbf{x}^s), g(\mathbf{x}^s)) + \mathbb{E}_{\mathbf{x}^t \in \mathbf{X}_T} k(g(\mathbf{x}^t), g(\mathbf{x}^t)) - 2\mathbb{E}_{\mathbf{x}^s \in \mathbf{X}_S} \mathbb{E}_{\mathbf{x}^t \in \mathbf{X}_T} k(g(\mathbf{x}^s), g(\mathbf{x}^t)).$$
A similar idea has been used in DAN (Long et al., 2015), JAN (Long et al., 2017).

3. **Central Moment Discrepancy (CMD).** Given moment $K$, a range $[a, b]^d$. Denote $C_k(\mathbf{X}) = \mathbb{E}_{\mathbf{x} \in \mathbf{X}}((\mathbf{x} - \mathbb{E}(\mathbf{x}))^k)$.
$$\Omega(\mathbf{X}_S, \mathbf{X}_T, g, K) = \frac{1}{|b - a|} \left\| \mathbb{E}_{\mathbf{x}^s \in \mathbf{X}_S}(g(\mathbf{x}^s)) - \mathbb{E}_{\mathbf{x}^t \in \mathbf{X}_T}(g(\mathbf{x}^t)) \right\|_2$$
$$+ \sum_{k=2}^{K} \frac{1}{|b - a|^k} \left\| C_k(g(\mathbf{X}_S)) - C_k(g(\mathbf{X}_T)) \right\|_2$$

4. **CORrelation ALignment (CORAL) (Sun & Saenko, 2016).** Define the covariance matrix $\mathrm{Cov}(\mathbf{X}) = \mathbb{E}_{\mathbf{x}_i \in \mathbf{X}, \mathbf{x}_j \in \mathbf{X}} \left[ (\mathbf{x}_i - \mathbb{E}[\mathbf{x}_i])(\mathbf{x}_j - \mathbb{E}[\mathbf{x}_j]) \right]$.
$$\Omega(\mathbf{X}_S, \mathbf{X}_T, g) = \left\| \mathrm{Cov}(g(\mathbf{X}_S)) - \mathrm{Cov}(g(\mathbf{X}_T)) \right\|_F^2.$$

5. **Kullback-Leibler divergence (KL) (Nguyen et al., 2021).**
$$\Omega(\mathbf{X}_S, \mathbf{X}_T, g)$$
$$= \mathbb{E}_{\mathbf{x}^t \in \mathbf{X}_T} \left[ \log(p_T(g(\mathbf{x}^t))) - \log(p_S(g(\mathbf{x}^t))) \right] + \mathbb{E}_{\mathbf{x}^s \in \mathbf{X}_S} \left[ \log(p_S(g(\mathbf{x}^s))) - \log(p_T(g(\mathbf{x}^s))) \right]$$
where $p_S(z) \approx \mathbb{E}_{\mathbf{x}^s \in \mathbf{X}_S} p(z|\mathbf{x}^s), p_T(z) \approx \mathbb{E}_{\mathbf{x}^t \in \mathbf{X}_T} p(z|\mathbf{x}^t)$, $p(z|\mathbf{x})$ is a Gaussian distribution with a diagonal covariance matrix and $z$ is sampled via reparameterization trick.

6. **Wasserstein Distance (WD) (Shen et al., 2018).** Hyperparameter $\lambda$.
$$\Omega(\mathbf{X}_S, \mathbf{X}_T, f, g, d) = \sup_d \left( \mathbb{E}_{\mathbf{x}^s \in \mathbf{X}_S}(d \circ g)(\mathbf{x}^s) - \mathbb{E}_{\mathbf{x}^t \in \mathbf{X}_T}(d \circ g)(\mathbf{x}^t) - \lambda \left( \nabla_{g(\mathbf{x})}(d \circ g)(\mathbf{x}) - 1 \right)^2 \right).$$

## C.3 ARCHITECTURES

For the DIGIT dataset, we use the same convolutional neural network that has been used in (Gulrajani & Lopez-Paz, 2020)– see Table 5 for details.

## C.4 DATA PREPROCESSING AND AUGMENTATION

For DIGIT datasets, we only resize all images to $32 \times 32$ pixels. For non-DIGIT datasets, we apply the following standard data augmentation techniques (Gulrajani & Lopez-Paz, 2020) (for both the labeled source training data and the unlabeled target training data): crops of random size and aspect ratio, resizing to $224 \times 224$ pixels, random horizontal flips, random color jitter, grayscaling the image with 10% probability; and normalized the data using ImageNet channel means and standard deviations. Note that for SRoUDA, we do not apply the proposed random masked augmentation (Zhu et al., 2023) as well as RandAugment (Cubuk et al., 2019) to ensure a fair comparison across all methods.

| # | Layer |
|---|---|
| 1 | Conv2D(in=$d$, out=64) |
| 2 | ReLU |
| 3 | GroupNorm(groups=8) |
| 4 | Conv2D(in=64,out=128,stride=2) |
| 5 | ReLU |
| 6 | GroupNorm(8 groups) |
| 7 | Conv2D(in=128,out=128) |
| 8 | ReLU |
| 9 | GroupNorm(8 groups) |
| 10 | Conv2D(in=128,out=128) |
| 11 | ReLU |
| 12 | GroupNorm(8 groups) |
| 13 | Global average-pooling |

Table 5: Details of Convolutional network architecture for DIGIT datasets (including MNIST, MNIST-M, SVHN, SYN, USPS). All convolutions use $3 \times 3$ kernels and "same" padding.

## C.5 HYPERPARAMETERS

| Condition | Parameter | Default value | Random distribution |
|---|---|---|---|
| Network | Resnet dropout rate | 0 | Uniform( [0, 0.1, 0.5]) |
| Algorithm | $\lambda_1$ | 1.0 | $10^{\text{Uniform}(-1,1)}$ |
| | $\lambda_2$ | 1.0 | $10^{\text{Uniform}(-1,1)}$ |
| | discriminator steps | 1 | $2^{\text{Uniform}(0,3)}$ |
| | adam $\beta_1$ | 0.9 | Uniform( 0,0.9) |
| DIGIT | data augmentation | False | False |
| | batch size | 128 | 128 |
| | number of iterations | 20k | 20k |
| | learning rate | 0.001 | $10^{\text{Uniform}(-4.5,-2.5)}$ |
| | discriminator learning rate | 0.001 | $10^{\text{Uniform}(-4.5,-2.5)}$ |
| | weight decay | 0 | 0 |
| | discriminator weight decay | 0 | $10^{\text{Uniform}(-6,-3)}$ |
| not DIGIT | data augmentation | True | True |
| | batch size | 16 | 16 |
| | number of iterations | 25k | 25k |
| | learning rate | 0.00005 | $10^{\text{Uniform}(-5,-3.5)}$ |
| | discriminator learning rate | 0.00005 | $10^{\text{Uniform}(-5,-3.5)}$ |
| | weight decay | 0.0001 | $10^{\text{Uniform}(-5,-2)}$ |
| | discriminator weight decay | 0.0001 | $10^{\text{Uniform}(-5,-2)}$ |

Table 6: Hyperparameters, the default values and distributions for random search.

# D RESULTS ON ALL SOURCE-TARGET PAIRS

## D.1 DIGITS

| Source→Target | SVHN→MNIST | | | SVHN→MNIST-M | | | SVHN→SYN | | | SVHN→USPS | | |
|---|---|---|---|---|---|---|---|---|---|---|---|---|
| Algorithm | clean acc | pgd acc | aa acc | clean acc | pgd acc | aa acc | clean acc | pgd acc | aa acc | clean acc | pgd acc | aa acc |
| Natural DANN | 82.6±0.2 | 64.9±0.9 | 64.0±0.9 | 51.8±0.6 | 21.0±2.3 | 20.2±2.4 | 93.8±0.1 | 80.2±0.4 | 79.2±0.4 | 88.0±1.0 | 69.2±1.1 | 67.4±1.6 |
| AT(src only) | 82.3±0.4 | 75.1±0.4 | 74.7±0.4 | 55.7±0.3 | 36.7±0.3 | 35.5±0.3 | 94.8±0.2 | 90.6±0.1 | 90.4±0.1 | 90.2±0.3 | 82.0±0.3 | 81.4±0.2 |
| TRADES(src only) | 83.0±0.6 | 74.9±0.2 | 74.4±0.2 | 54.3±0.3 | 38.9±0.2 | 37.6±0.2 | 94.3±0.1 | 90.7±0.1 | 90.4±0.2 | 91.5±0.3 | 81.7±0.1 | 80.9±0.1 |
| AT(tgt,pseudo) | 87.5±0.1 | 85.8±0.1 | 85.8±0.1 | 59.2±0.2 | 47.0±0.3 | 46.0±0.3 | 95.6±2.0 | 92.4±0.1 | 92.3±0.1 | 93.8±0.5 | 91.7±0.4 | 91.6±0.3 |
| TRADES(tgt,pseudo) | 86.6±0.2 | 84.8±0.1 | 84.7±0.1 | 61.1±0.3 | 50.4±0.6 | 49.4±0.5 | 95.7±0.2 | 93.3±0.2 | 93.2±0.2 | 94.3±0.4 | 92.1±0.2 | 92.0±0.1 |
| AT+UDA | 81.6±0.6 | 71.7±0.5 | 70.9±0.6 | 55.8±0.3 | 39.5±0.2 | 38.5±0.2 | 94.8±0.0 | 90.7±0.2 | 90.4±0.1 | 88.4±1.2 | 80.2±1.1 | 79.4±1.2 |
| ARTUDA | 92.6±0.7 | 91.2±0.7 | 91.1±0.7 | 58.0±0.7 | 48.0±1.2 | 47.2±1.3 | 97.0±0.2 | 94.8±0.1 | 94.7±0.1 | 98.1±0.1 | 97.0±0.2 | 96.9±0.1 |
| SROUDA | 86.5±0.1 | 84.0±0.2 | 83.9±0.2 | 59.9±0.6 | 50.1±0.6 | 48.9±0.6 | 95.9±0.1 | 93.9±0.1 | 93.8±0.1 | 94.5±0.3 | 91.9±0.9 | 91.8±0.9 |
| DART(clean src) | **98.7±0.1** | 98.2±0.2 | 98.2±0.2 | 70.2±1.4 | 61.7±1.3 | 61.4±1.3 | **97.2±0.1** | 94.4±0.3 | 94.3±0.3 | **98.5±0.1** | **97.8±0.1** | **97.7±0.1** |
| DART(adv src) | **98.7±0.1** | **98.3±0.2** | **98.3±0.2** | 68.5±1.5 | 59.8±1.2 | 59.3±1.2 | 97.0±0.0 | **95.0±0.0** | **94.9±0.0** | 98.3±0.4 | 97.5±0.4 | 97.4±0.4 |
| DART(kl src) | 98.6±0.2 | 98.2±0.2 | 98.2±0.2 | **72.6±1.4** | **63.6±1.7** | **63.2±1.8** | 97.1±0.1 | 94.9±0.1 | 94.8±0.1 | 98.4±0.0 | 97.4±0.1 | **97.7±0.1** |

Table 7: Standard accuracy (nat acc) / Robust accuracy under PGD attack (pgd acc)/ Robust accuracy under AutoAttack (aa acc) of target test data on DIGIT dataset with a fix source domain SVHN and different target domains.

| Source→Target | SYN→MNIST | | | SYN→MNIST-M | | | SYN→SVHN | | | SYN→USPS | | |
|---|---|---|---|---|---|---|---|---|---|---|---|---|
| Algorithm | clean acc | pgd acc | aa acc | clean acc | pgd acc | aa acc | clean acc | pgd acc | aa acc | clean acc | pgd acc | aa acc |
| Natural DANN | 96.3±0.2 | 89.0±1.3 | 88.6±1.4 | 60.4±0.8 | 38.7±0.3 | 38.3±0.3 | 82.2±0.7 | 37.9±0.7 | 36.3±0.6 | 97.5±0.1 | 85.6±0.7 | 85.1±0.8 |
| AT(src only) | 96.6±0.1 | 94.4±0.1 | 94.3±0.1 | 63.4±0.5 | 43.1±0.3 | 42.9±0.3 | 79.0±0.5 | 49.2±0.3 | 48.4±0.3 | 97.3±0.0 | 93.4±0.1 | 93.2±0.1 |
| TRADES(src only) | 96.5±0.0 | 94.3±0.1 | 94.2±0.1 | 63.1±0.2 | 43.8±0.4 | 43.5±0.4 | 76.7±0.5 | 52.4±0.1 | 51.1±0.3 | 97.1±0.2 | 93.4±0.1 | 93.2±0.1 |
| AT(tgt,pseudo) | 97.2±0.0 | 96.8±0.0 | 96.8±0.0 | 65.9±0.6 | 55.8±0.5 | 55.2±0.5 | 84.6±0.2 | 70.9±0.0 | 69.9±0.1 | 98.1±0.1 | 97.3±0.2 | 97.3±0.2 |
| TRADES(tgt,pseudo) | 97.3±0.0 | 96.9±0.0 | 96.9±0.0 | 66.5±0.3 | 58.6±0.6 | 58.0±0.6 | 84.7±0.3 | **73.8±0.2** | **72.4±0.2** | 98.3±0.1 | 97.2±0.1 | 97.2±0.1 |
| AT+UDA | 95.9±0.1 | 94.0±0.1 | 94.0±0.1 | 68.2±0.4 | 51.5±0.2 | 51.3±0.2 | 82.3±0.3 | 53.5±0.3 | 52.8±0.3 | 96.2±0.2 | 92.7±0.2 | 92.5±0.2 |
| ARTUDA | **98.6±0.1** | **98.2±0.1** | **98.2±0.1** | 69.6±0.6 | 61.6±0.7 | 60.9±0.4 | 83.4±0.6 | 69.4±1.2 | 68.3±1.4 | **98.6±0.1** | **97.8±0.1** | **97.8±0.1** |
| SROUDA | 97.0±0.0 | 96.0±0.1 | 96.0±0.1 | 63.6±0.4 | 55.0±0.1 | 54.4±0.1 | 84.8±0.2 | 71.1±0.1 | 69.6±0.1 | 98.1±0.1 | 96.9±0.3 | 96.8±0.4 |
| DART(clean src) | 98.3±0.3 | 97.9±0.3 | 97.9±0.3 | **75.2±0.8** | **66.7±0.8** | **66.5±0.8** | **86.2±0.1** | 72.8±0.3 | 72.2±0.3 | 98.6±0.2 | 97.8±0.3 | 97.8±0.3 |
| DART(adv src) | 98.1±0.3 | 97.5±0.2 | 97.5±0.2 | 72.6±0.9 | 64.3±1.0 | 64.1±1.0 | 86.1±0.1 | 73.0±0.2 | 72.3±0.2 | 98.4±0.1 | 97.6±0.1 | 97.6±0.1 |
| DART(kl src) | 98.2±0.3 | 97.8±0.3 | 97.8±0.3 | 74.4±1.2 | 66.0±1.3 | 65.8±1.3 | 86.2±0.2 | 72.8±0.3 | 72.1±0.3 | 98.4±0.1 | 97.5±0.0 | 97.5±0.0 |

Table 8: Standard / Robust accuracy (%) of target test data on DIGIT dataset with a fix source domain SYN and different target domains.

| Source→Target | USPS→MNIST | | | USPS→MNIST-M | | | USPS→SVHN | | | USPS→SYN | | |
|---|---|---|---|---|---|---|---|---|---|---|---|---|
| Algorithm | clean acc | pgd acc | aa acc | clean acc | pgd acc | aa acc | clean acc | pgd acc | aa acc | clean acc | pgd acc | aa acc |
| Natural DANN | 98.3±0.1 | 95.9±0.4 | 95.8±0.4 | 54.1±3.4 | 40.7±2.2 | 40.5±2.2 | 21.2±1.4 | 9.8±1.4 | 9.6±1.4 | 40.8±1.6 | 33.2±1.8 | 33.1±1.8 |
| AT(src only) | 98.3±0.0 | 97.5±0.0 | 97.5±0.0 | 60.7±0.2 | 48.2±0.2 | 48.0±0.2 | 24.0±0.3 | 15.5±0.2 | 15.3±0.2 | 46.3±0.4 | 38.6±0.4 | 38.5±0.4 |
| TRADES(src only) | 98.4±0.0 | 97.5±0.0 | 97.5±0.0 | 60.7±0.3 | 48.3±0.1 | 48.1±0.1 | 24.4±0.2 | 15.1±0.3 | 14.9±0.3 | 46.2±0.4 | 39.1±0.4 | 39.0±0.4 |
| AT(tgt,pseudo) | 98.5±0.1 | 98.2±0.1 | 98.2±0.1 | 63.9±0.3 | 56.3±0.1 | 56.0±0.1 | 24.9±0.1 | 19.7±0.1 | 19.5±0.1 | 45.5±0.1 | 41.9±0.2 | 41.8±0.2 |
| TRADES(tgt,pseudo) | 98.5±0.1 | 98.2±0.1 | 98.2±0.1 | 64.1±0.0 | 58.3±0.2 | 57.9±0.2 | 24.7±0.1 | 20.3±0.2 | 20.1±0.2 | 45.3±0.2 | 42.6±0.2 | 42.5±0.2 |
| AT+UDA | 98.2±0.0 | 97.6±0.1 | 97.6±0.1 | 62.4±0.5 | 50.4±0.4 | 50.1±0.3 | 23.0±0.1 | 14.6±1.0 | 14.5±1.1 | 45.7±1.1 | 39.7±0.5 | 39.6±0.4 |
| ARTUDA | **99.0±0.0** | **98.8±0.0** | **98.8±0.0** | 56.9±0.7 | 52.2±0.8 | 52.1±0.8 | 23.9±0.9 | 20.0±0.3 | 19.9±0.3 | 49.1±1.1 | 49.3±0.5 | 49.3±0.5 |
| SROUDA | 98.4±0.0 | 97.9±0.1 | 97.9±0.1 | 62.5±0.1 | 54.2±0.4 | 53.7±0.4 | 21.4±0.8 | 18.0±0.3 | 17.9±0.3 | 45.6±0.0 | 41.6±0.1 | 41.4±0.1 |
| DART(clean src) | 98.8±0.0 | 98.4±0.0 | 98.4±0.0 | 66.8±1.0 | 60.7±0.8 | 60.6±0.8 | 29.1±0.4 | 25.2±0.2 | 25.1±0.2 | 53.2±0.5 | 50.7±0.4 | 50.6±0.4 |
| DART(adv src) | 98.8±0.0 | 98.5±0.0 | 98.5±0.0 | 67.3±0.8 | 61.0±0.9 | 60.8±0.9 | **29.7±0.8** | **25.5±0.5** | **25.4±0.4** | 53.0±0.2 | 50.6±0.2 | 50.6±0.2 |
| DART(kl src) | 98.8±0.1 | 98.5±0.1 | 98.5±0.1 | **68.4±0.8** | **62.0±0.8** | **61.8±0.8** | 29.0±0.9 | 25.2±0.7 | 25.1±0.7 | **53.8±0.5** | **51.3±0.6** | **51.3±0.6** |

Table 9: Standard / Robust accuracy (%) of target test data on DIGIT dataset with a fix source domain USPS and different target domains.

| Source→Target | MNIST→MNIST-M | | | MNIST→SVHN | | | MNIST→SYN | | | MNIST→USPS | | |
|---|---|---|---|---|---|---|---|---|---|---|---|---|
| Algorithm | clean acc | pgd acc | aa acc | clean acc | pgd acc | aa acc | clean acc | pgd acc | aa acc | clean acc | pgd acc | aa acc |
| Natural DANN | 62.4±3.9 | 45.9±3.2 | 45.6±3.2 | 21.8±0.2 | 12.4±0.6 | 12.2±0.7 | 47.0±1.4 | 39.0±2.5 | 38.8±2.5 | 98.8±0.2 | 97.2±0.4 | 97.1±0.4 |
| AT(src only) | 67.7±0.5 | 51.8±0.0 | 51.4±0.0 | 25.5±1.2 | 14.4±0.1 | 14.2±0.0 | 50.0±0.5 | 44.1±0.4 | 44.0±0.4 | 99.0±0.1 | 98.1±0.1 | 98.1±0.1 |
| TRADES(src only) | 67.0±0.4 | 52.2±0.2 | 51.9±0.2 | **25.5±0.9** | 14.1±0.1 | 13.9±0.1 | 49.6±0.7 | 43.9±0.5 | 43.8±0.5 | 98.8±0.2 | 98.0±0.1 | 98.0±0.1 |
| AT(tgt,pseudo) | 70.2±0.1 | 61.8±0.3 | 61.4±0.3 | 22.9±0.0 | 17.9±0.1 | 17.7±0.1 | 51.0±0.5 | 48.0±0.5 | 47.9±0.5 | 99.0±0.2 | 98.4±0.2 | 98.4±0.2 |
| TRADES(tgt,pseudo) | 70.0±0.2 | 63.7±0.1 | 63.2±0.2 | 22.5±0.1 | 17.9±0.2 | 17.7±0.2 | 51.2±0.6 | 48.9±0.7 | 48.8±0.7 | 99.1±0.2 | 98.6±0.2 | 98.6±0.2 |
| AT+UDA | 68.9±0.5 | 54.7±0.4 | 54.2±0.3 | 21.3±1.3 | 17.6±0.5 | 17.5±0.5 | 49.9±0.4 | 44.0±0.4 | 43.8±0.4 | 98.9±0.1 | 98.1±0.1 | 98.1±0.1 |
| ARTUDA | 63.3±0.4 | 56.5±0.7 | 56.3±0.7 | 20.0±0.4 | 18.7±0.1 | 18.6±0.1 | 52.2±0.5 | 51.0±0.6 | 50.9±0.6 | **99.2±0.1** | **98.8±0.2** | **98.8±0.2** |
| SROUDA | 69.9±0.2 | 62.0±0.2 | 61.4±0.2 | 23.0±1.8 | 18.8±0.4 | 18.7±0.3 | 51.2±0.5 | 48.6±0.6 | 48.5±0.6 | 99.0±0.2 | 98.5±0.2 | 98.5±0.2 |
| DART(clean src) | 77.7±1.8 | 71.2±1.9 | 71.1±1.9 | 22.5±0.1 | 19.1±0.3 | 19.0±0.3 | 53.2±0.6 | 51.1±0.6 | 51.1±0.6 | 99.1±0.1 | 98.5±0.1 | 98.5±0.1 |
| DART(adv src) | **78.4±0.3** | **71.3±0.2** | **71.1±0.2** | 22.6±0.3 | 19.4±0.1 | 19.3±0.1 | 53.2±0.5 | 51.2±0.6 | 51.2±0.6 | 99.1±0.2 | 98.4±0.1 | 98.4±0.1 |
| DART(kl src) | 77.3±2.1 | 70.6±2.1 | 70.4±2.1 | 22.5±0.1 | **19.8±0.2** | **19.7±0.2** | **53.4±0.3** | **51.4±0.2** | **51.3±0.2** | 99.1±0.1 | 98.6±0.1 | 98.6±0.1 |

Table 10: Standard accuracy (nat acc) / Robust accuracy under PGD attack (pgd acc)/ Robust accuracy under AutoAttack (aa acc) of DIGIT dataset with a fix source domain MNIST and different target domains.

| Source→Target | MNIST-M→MNIST | | | MNIST-M→SVHN | | | MNIST-M→SYN | | | MNIST-M→USPS | | |
|---|---|---|---|---|---|---|---|---|---|---|---|---|
| Algorithm | clean acc | pgd acc | aa acc | clean acc | pgd acc | aa acc | clean acc | pgd acc | aa acc | clean acc | pgd acc | aa acc |
| Natural DANN | 98.4±0.1 | 94.1±1.1 | 94.0±1.2 | 35.9±0.8 | 5.7±1.6 | 5.4±1.5 | 69.0±1.4 | 38.4±1.2 | 38.0±1.2 | 97.5±0.1 | 78.9±2.8 | 78.1±3.1 |
| AT(src only) | 98.7±0.1 | 97.8±0.1 | 97.7±0.1 | 34.0±0.5 | 22.8±0.2 | 22.3±0.2 | 68.9±0.1 | 54.8±0.3 | 54.4±0.3 | 96.8±0.1 | 91.2±0.4 | 91.0±0.5 |
| TRADES(src only) | 98.6±0.1 | 97.7±0.0 | 97.7±0.0 | 32.0±0.1 | 24.3±0.1 | 23.8±0.1 | 67.3±0.6 | 54.9±0.2 | 54.3±0.2 | 96.6±0.4 | 92.3±0.3 | 92.2±0.3 |
| AT(tgt,pseudo) | 98.9±0.1 | 98.6±0.0 | 98.6±0.0 | 43.4±0.1 | 32.0±0.4 | 30.8±0.4 | 77.3±0.3 | 70.1±0.4 | 69.8±0.4 | 98.3±0.1 | 97.5±0.1 | 97.5±0.1 |
| TRADES(tgt,pseudo) | 98.9±0.1 | 98.6±0.1 | 98.6±0.1 | 43.2±0.5 | 31.7±0.4 | 30.4±0.4 | 77.7±0.3 | 72.0±0.6 | 71.7±0.6 | 98.5±0.3 | 97.6±0.2 | 97.6±0.2 |
| AT+UDA | 98.8±0.1 | 98.0±0.1 | 98.0±0.1 | 37.3±1.2 | 18.2±0.8 | 17.5±0.8 | 73.5±0.6 | 61.6±0.5 | 61.3±0.6 | 96.4±0.5 | 92.1±0.8 | 91.9±0.8 |
| ARTUDA | 99.2±0.1 | 99.0±0.1 | 99.0±0.1 | 34.5±2.9 | 22.6±0.8 | 21.4±0.7 | 92.3±0.8 | **88.5±1.3** | 88.2±1.3 | 99.0±0.1 | 98.1±0.3 | 98.1±0.2 |
| SROUDA | 99.1±0.0 | 98.9±0.0 | 98.9±0.0 | 48.2±0.4 | 38.2±0.1 | 37.1±0.2 | 76.4±0.1 | 70.4±0.2 | 70.1±0.2 | 98.4±0.1 | 97.7±0.2 | 97.7±0.2 |
| DART(clean src) | 99.3±0.0 | 98.9±0.0 | 98.9±0.0 | **52.3±2.4** | **41.7±1.6** | **41.3±1.6** | **92.6±0.4** | 88.3±0.8 | **88.2±0.9** | 99.1±0.1 | 98.2±0.2 | 98.2±0.2 |
| DART(adv src) | **99.4±0.1** | 99.1±0.1 | 99.1±0.1 | 48.4±1.8 | 38.6±1.3 | 38.2±1.3 | 89.6±0.9 | 85.5±1.1 | 85.4±1.1 | 98.9±0.1 | 98.2±0.1 | 98.2±0.1 |
| DART(kl src) | 99.3±0.0 | **99.1±0.0** | **99.1±0.0** | 49.9±1.8 | 40.3±1.2 | 40.0±1.2 | 91.8±0.8 | 87.4±1.1 | 87.2±1.1 | 99.0±0.2 | **98.5±0.2** | **98.5±0.2** |

Table 11: Standard accuracy (nat acc) / Robust accuracy under PGD attack (pgd acc)/ Robust accuracy under AutoAttack (aa acc) of target test data on DIGIT dataset with a fix source domain MNIST-M and different target domains.

## D.2 OFFICEHOME

| Source→Target | RealWorld→Art | | | RealWorld→Clipart | | | RealWorld→Product | | |
|---|---|---|---|---|---|---|---|---|---|
| Algorithm | clean acc | pgd acc | aa acc | clean acc | pgd acc | aa acc | clean acc | pgd acc | aa acc |
| Natural DANN | **61.0±0.6** | 0.4±0.1 | 0.0±0.0 | 55.5±0.6 | 3.8±0.5 | 1.1±0.2 | **74.3±0.9** | 1.9±0.2 | 0.2±0.1 |
| AT(src only) | 47.2±1.2 | 28.0±1.2 | 27.3±1.1 | 54.1±0.8 | 40.9±1.0 | 39.7±1.0 | 66.7±0.3 | 49.9±1.1 | 48.9±1.3 |
| TRADES(src only) | 45.6±0.4 | 27.5±1.3 | 26.5±1.4 | 53.9±1.7 | 41.9±0.7 | 41.0±0.6 | 66.9±1.3 | 48.9±0.8 | 47.4±0.5 |
| AT(tgt,pseudo) | 46.4±0.8 | 29.4±1.4 | 28.8±1.2 | 55.0±0.5 | 49.4±0.6 | 48.9±0.5 | 72.3±1.5 | 60.4±0.8 | 59.6±0.9 |
| TRADES(tgt,pseudo) | 47.9±2.4 | 27.4±0.4 | 26.5±0.3 | 55.6±0.9 | 49.7±0.8 | 49.3±0.8 | 70.0±1.4 | 61.9±1.2 | 61.3±1.2 |
| AT+UDA | 50.3±1.5 | 27.7±0.4 | 26.5±0.3 | 53.8±1.4 | 44.2±0.3 | 43.3±0.2 | 67.0±1.0 | 51.2±0.9 | 49.7±0.9 |
| ARTUDA | 49.5±2.0 | 28.4±1.0 | 26.1±1.1 | 58.3±0.7 | 48.5±0.9 | 46.7±0.5 | 73.0±0.6 | 58.3±0.4 | 55.7±0.6 |
| SROUDA | 42.1±1.4 | 27.5±0.1 | 25.2±0.3 | 55.4±0.6 | 47.3±0.7 | 46.2±0.8 | 70.7±0.6 | 60.6±1.2 | 58.6±2.0 |
| DART(clean src) | 53.7±1.0 | 29.1±1.6 | 27.2±1.1 | **58.6±0.4** | 49.8±1.1 | 49.0±1.0 | 74.0±0.9 | 60.2±1.5 | 59.1±1.4 |
| DART(adv src) | 49.8±2.3 | **32.3±1.7** | **31.3±1.4** | 57.8±0.5 | **52.5±0.5** | **51.9±0.5** | 73.8±0.7 | 63.1±0.2 | 62.3±0.2 |
| DART(kl src) | 53.4±1.5 | 32.0±1.6 | 30.8±1.6 | 57.4±0.5 | 51.8±0.9 | 51.0±0.8 | 73.0±0.6 | **63.2±0.9** | **62.5±0.9** |

Table 12: Standard accuracy (nat acc) / Robust accuracy under PGD attack (pgd acc)/ Robust accuracy under AutoAttack (aa acc) of target test data on OfficeHome dataset with a fix source domain RealWorld and different target domains.

| Source→Target | Art→Clipart | | | Art→Product | | | Art→RealWorld | | |
|---|---|---|---|---|---|---|---|---|---|
| Algorithm | clean acc | pgd acc | aa acc | clean acc | pgd acc | aa acc | clean acc | pgd acc | aa acc |
| Natural DANN | 49.1±0.3 | 2.8±0.4 | 1.1±0.2 | 55.5±0.8 | 0.9±0.2 | 0.3±0.1 | **66.8±0.8** | 1.0±0.3 | 0.1±0.1 |
| AT(src only) | 45.4±0.6 | 32.0±0.3 | 30.7±0.2 | 48.5±0.4 | 29.8±0.3 | 28.0±0.6 | 57.2±2.1 | 36.1±0.8 | 34.7±1.1 |
| TRADES(src only) | 46.1±0.7 | 32.8±0.6 | 31.5±0.8 | 50.4±0.6 | 31.2±0.1 | 29.5±0.1 | 58.8±1.3 | 35.2±0.6 | 33.4±0.7 |
| AT(tgt,pseudo) | 48.0±0.5 | 41.7±0.7 | 41.2±0.7 | 55.9±0.4 | 46.2±0.3 | 45.6±0.5 | 57.6±0.6 | 40.5±1.1 | 39.6±1.0 |
| TRADES(tgt,pseudo) | 48.6±0.3 | 43.6±0.4 | 43.1±0.4 | 55.9±0.4 | 47.6±0.1 | **47.2±0.3** | 57.1±1.6 | 41.8±0.2 | 40.6±0.5 |
| AT+UDA | 45.6±0.6 | 32.9±0.6 | 32.2±0.5 | 48.4±1.0 | 30.0±1.0 | 28.6±1.2 | 56.2±1.6 | 34.6±0.9 | 33.3±0.8 |
| ARTUDA | 50.9±1.6 | 41.7±1.7 | 40.0±2.0 | 55.0±0.8 | 41.2±1.0 | 39.2±1.4 | 61.7±0.6 | 42.5±1.0 | 39.6±0.4 |
| SROUDA | 48.2±0.5 | 38.9±0.5 | 37.5±0.8 | 52.9±0.6 | 45.8±0.3 | 44.6±0.3 | 57.4±1.4 | **44.2±0.7** | **42.0±1.1** |
| DART(clean src) | 50.4±0.9 | 42.2±0.6 | 41.4±0.5 | **60.1±0.2** | 47.7±1.0 | 46.4±1.4 | 62.7±0.5 | 40.7±0.5 | 38.5±0.8 |
| DART(adv src) | 49.8±0.3 | 42.5±0.5 | 41.9±0.6 | 58.5±0.9 | 47.1±1.4 | 46.4±1.5 | 61.6±0.7 | 41.8±0.3 | 39.4±0.3 |
| DART(kl src) | **50.8±0.1** | **43.9±0.2** | **43.3±0.2** | 57.9±1.0 | 47.7±0.6 | 46.7±0.6 | 62.1±0.6 | 43.8±1.1 | 41.4±1.3 |

Table 13: Standard accuracy (nat acc) / Robust accuracy under PGD attack (pgd acc)/ Robust accuracy under AutoAttack (aa acc) of target test data on OfficeHome dataset with a fix source domain Art and different target domains.

| Source→Target | Clipart→Art | | | Clipart→Product | | | Clipart→RealWorld | | |
|---|---|---|---|---|---|---|---|---|---|
| Algorithm | clean acc | pgd acc | aa acc | clean acc | pgd acc | aa acc | clean acc | pgd acc | aa acc |
| Natural DANN | **45.2±0.8** | 0.0±0.0 | 0.0±0.0 | 47.9±0.8 | 3.6±1.0 | 1.1±0.3 | **67.4±1.5** | 0.6±0.3 | 0.1±0.1 |
| AT(src only) | 34.4±1.8 | 14.5±0.6 | 13.0±0.3 | 51.2±1.5 | 33.1±0.8 | 31.7±0.8 | 53.8±1.0 | 28.3±0.1 | 26.5±0.7 |
| TRADES(src only) | 30.6±2.5 | 16.6±0.4 | 15.1±0.5 | 48.6±1.5 | 34.1±0.9 | 32.8±0.7 | 50.2±2.1 | 30.9±0.8 | 28.7±1.2 |
| AT(tgt,pseudo) | 39.4±1.5 | 23.0±0.1 | 22.0±0.4 | 55.6±0.8 | 46.8±1.2 | 46.5±1.2 | 56.5±0.8 | 41.5±0.4 | 40.4±0.4 |
| TRADES(tgt,pseudo) | 40.0±1.0 | 22.0±0.4 | 21.1±0.5 | 56.2±0.3 | 47.9±0.5 | 47.3±0.4 | 56.2±0.3 | **43.8±1.0** | **42.8±0.5** |
| AT+UDA | 39.6±1.9 | 16.4±1.0 | 15.2±1.2 | 52.4±1.1 | 34.5±0.7 | 32.5±1.3 | 57.6±0.5 | 32.0±0.8 | 28.0±1.8 |
| ARTUDA | 42.0±0.2 | 20.2±1.0 | 18.9±1.2 | 56.1±1.3 | 44.1±1.4 | 42.9±1.5 | 58.9±1.2 | 39.2±0.6 | 37.9±0.5 |
| SROUDA | 36.3±0.3 | 23.8±0.6 | 21.3±0.1 | 53.9±1.0 | 47.2±1.1 | 45.7±1.2 | 55.1±1.7 | 42.1±0.8 | 39.9±1.1 |
| DART(clean src) | 44.1±0.9 | 24.2±0.5 | 22.6±0.3 | 57.0±0.3 | 45.5±0.6 | 44.8±0.5 | 57.8±0.3 | 39.6±0.2 | 38.3±0.3 |
| DART(adv src) | 43.0±1.3 | **26.1±1.1** | **25.0±1.0** | 58.0±1.0 | 47.6±0.9 | 47.0±0.8 | 58.0±0.2 | 41.5±0.9 | 40.4±0.8 |
| DART(kl src) | 42.6±0.8 | 24.6±0.8 | 23.0±0.7 | **58.3±0.8** | **48.8±1.4** | **48.5±1.3** | 58.9±0.8 | 40.8±0.6 | 39.9±0.4 |

Table 14: Standard accuracy (nat acc) / Robust accuracy under PGD attack (pgd acc)/ Robust accuracy under AutoAttack (aa acc) of target test data on OfficeHome dataset with a fix source domain Clipart and different target domains.

| Source→Target | Product→Art | | | Product→Clipart | | | Product→RealWorld | | |
|---|---|---|---|---|---|---|---|---|---|
| Algorithm | clean acc | pgd acc | aa acc | clean acc | pgd acc | aa acc | clean acc | pgd acc | aa acc |
| Natural DANN | 49.1±0.3 | 0.2±0.1 | 0.0±0.0 | **57.4±0.2** | 2.0±0.5 | 0.3±0.1 | 60.0±0.6 | 0.3±0.1 | 0.0±0.0 |
| AT(src only) | 33.8±1.3 | 15.3±0.4 | 13.8±0.4 | 47.2±0.1 | 34.1±0.6 | 32.1±0.6 | 56.9±1.3 | 34.6±1.0 | 32.1±0.9 |
| TRADES(src only) | 29.5±3.1 | 13.5±0.9 | 12.5±0.9 | 45.7±1.1 | 32.0±0.4 | 30.9±0.4 | 54.5±0.6 | 33.4±0.1 | 32.1±0.1 |
| AT(tgt,pseudo) | 38.5±1.6 | 20.3±0.6 | 19.3±0.6 | 49.1±0.8 | 42.9±0.4 | 42.3±0.6 | 61.4±1.5 | 44.2±1.2 | 43.3±1.2 |
| TRADES(tgt,pseudo) | 37.7±2.2 | 22.1±0.8 | 21.2±0.9 | 49.3±1.1 | 44.3±1.7 | 43.8±1.9 | 61.6±0.9 | 44.0±1.5 | 42.9±1.4 |
| AT+UDA | 36.1±3.4 | 14.8±0.9 | 14.2±0.6 | 48.9±0.7 | 37.9±1.3 | 36.7±1.4 | 59.3±1.8 | 35.8±1.1 | 34.4±1.2 |
| ARTUDA | 38.3±2.1 | 18.0±1.4 | 15.8±1.1 | 48.5±0.9 | 42.8±0.8 | 42.2±0.6 | 62.4±0.3 | 42.7±2.0 | 40.9±2.3 |
| SROUDA | 33.5±1.3 | 22.4±1.3 | 20.8±1.1 | 49.9±0.4 | 41.6±0.6 | 39.9±0.3 | 60.2±2.0 | 45.6±0.7 | 43.2±0.7 |
| DART(clean src) | **43.7±2.5** | 21.5±0.8 | 20.0±1.0 | 52.5±1.3 | 44.8±1.3 | 43.7±1.4 | 63.5±0.8 | 43.6±0.5 | 42.6±0.5 |
| DART(adv src) | 41.7±0.5 | **23.9±0.5** | **22.2±0.5** | 50.0±0.7 | **44.8±0.9** | **44.4±0.9** | **64.4±1.1** | **47.7±0.9** | **46.4±1.0** |
| DART(kl src) | 41.8±1.0 | 23.0±0.0 | 21.0±0.1 | 52.0±0.9 | 44.3±1.1 | 43.6±1.2 | 64.2±1.6 | 44.5±1.2 | 43.3±1.2 |

Table 15: Standard accuracy (nat acc) / Robust accuracy under PGD attack (pgd acc)/ Robust accuracy under AutoAttack (aa acc) of target test data on OfficeHome dataset with a fix source domain Product and different target domains.

## D.3 PACS

| Source→Target | Photo→Art | | | Photo→Clipart | | | Photo→Sketch | | |
|---|---|---|---|---|---|---|---|---|---|
| Algorithm | clean acc | pgd acc | aa acc | clean acc | pgd acc | aa acc | clean acc | pgd acc | aa acc |
| Natural DANN | **89.1±0.2** | 3.9±3.1 | 0.0±0.0 | 80.5±0.2 | 11.5±2.5 | 2.2±1.3 | 74.0±1.1 | 24.0±2.1 | 5.6±1.3 |
| AT(src only) | 71.0±3.0 | 32.7±1.6 | 31.1±1.7 | 71.4±1.9 | 50.4±0.6 | 48.6±0.2 | 69.3±0.5 | 61.0±0.6 | 59.6±0.7 |
| TRADES(src only) | 61.5±2.8 | 33.2±0.7 | 32.4±0.5 | 72.9±2.2 | 50.0±0.7 | 47.3±0.5 | 68.9±0.8 | 59.3±1.0 | 58.1±1.4 |
| AT(tgt,pseudo) | 82.3±0.7 | 59.1±0.7 | 58.5±0.9 | 85.5±0.6 | 77.4±1.0 | 77.1±0.9 | 78.2±0.4 | 75.5±0.3 | 75.1±0.3 |
| TRADES(tgt,pseudo) | 82.1±1.0 | **63.2±1.5** | **62.1±1.3** | 84.4±0.2 | 76.7±0.9 | 76.5±0.8 | 78.7±0.5 | 75.3±0.7 | 74.9±0.7 |
| AT+UDA | 73.3±3.5 | 44.1±1.4 | 29.5±2.1 | 70.6±1.6 | 62.2±1.5 | 60.9±1.5 | 70.6±1.6 | 62.2±1.5 | 60.9±1.5 |
| ARTUDA | 85.9±1.1 | 60.1±1.4 | 56.3±1.2 | 87.5±1.7 | 78.1±0.5 | 77.5±0.6 | 74.9±1.3 | 70.4±1.4 | 69.2±1.3 |
| SROUDA | 76.1±1.7 | 56.4±0.2 | 54.7±0.3 | 82.4±1.3 | 71.7±1.8 | 70.1±1.8 | 71.9±0.9 | 63.7±1.2 | 60.8±2.0 |
| DART(clean src) | 85.2±1.2 | 58.0±0.9 | 56.7±1.3 | **89.4±0.8** | 80.5±0.3 | 79.9±0.1 | **82.5±0.8** | **79.9±0.4** | **79.5±0.5** |
| DART(adv src) | 84.1±1.2 | 59.3±0.3 | 58.5±0.2 | 87.7±0.7 | **80.7±0.5** | **80.1±0.4** | 81.0±1.0 | 78.1±0.4 | 77.7±0.4 |
| DART(kl src) | 84.1±0.4 | 58.8±1.5 | 57.8±1.5 | 87.3±0.4 | 79.5±0.8 | 79.3±0.8 | 82.4±1.6 | 79.2±1.2 | 78.8±1.1 |

Table 16: Standard accuracy (nat acc) / Robust accuracy under PGD attack (pgd acc)/ Robust accuracy under AutoAttack (aa acc) of target test data on PACS dataset with a fix source domain Photo and different target domains.

| Source→Target | Clipart→Art | | | Clipart→Photo | | | Clipart→Sketch | | |
|---|---|---|---|---|---|---|---|---|---|
| Algorithm | clean acc | pgd acc | aa acc | clean acc | pgd acc | aa acc | clean acc | pgd acc | aa acc |
| Natural DANN | **84.9±0.7** | 0.6±0.3 | 0.0±0.0 | 92.5±0.7 | 1.4±0.4 | 0.0±0.0 | 78.2±0.9 | 25.7±2.2 | 8.7±0.9 |
| AT(src only) | 59.6±1.0 | 30.2±1.0 | 28.9±1.0 | 77.9±1.9 | 53.8±0.5 | 51.8±0.3 | 77.1±0.9 | 66.6±0.6 | 65.1±0.8 |
| TRADES(src only) | 58.7±2.5 | 28.0±0.4 | 27.1±0.5 | 78.9±1.3 | 53.8±0.8 | 51.9±1.0 | 74.6±0.9 | 67.6±0.2 | 66.7±0.2 |
| AT(tgt,pseudo) | 76.2±1.7 | 55.0±1.6 | 54.7±1.6 | 93.3±0.5 | 80.3±0.8 | 80.0±0.8 | 80.0±0.3 | 77.4±0.2 | 77.1±0.3 |
| TRADES(tgt,pseudo) | 78.5±1.7 | **58.0±1.4** | **56.8±1.0** | 92.2±0.1 | **82.1±0.5** | **81.7±0.6** | 79.9±0.5 | 77.6±0.4 | 77.5±0.4 |
| AT+UDA | 68.9±1.2 | 46.2±5.9 | 23.3±1.7 | 78.8±2.3 | 61.3±2.0 | 41.8±5.1 | 75.9±1.7 | 67.7±1.4 | 66.8±1.2 |
| ARTUDA | 76.5±2.5 | 53.3±1.6 | 52.2±1.7 | 89.4±0.9 | 75.0±1.7 | 71.7±1.0 | 80.3±0.4 | 74.9±1.0 | 73.8±1.1 |
| SROUDA | 72.0±1.5 | 50.9±1.1 | 49.2±1.6 | 90.3±0.9 | 79.9±2.0 | 79.2±1.9 | 76.7±1.2 | 72.3±1.3 | 71.3±1.3 |
| DART(clean src) | 77.4±1.1 | 54.6±0.3 | 53.8±0.2 | **94.2±0.5** | 79.8±1.2 | 78.6±1.2 | 84.9±0.4 | 81.0±0.7 | 80.6±0.7 |
| DART(adv src) | 78.2±1.3 | 56.3±1.3 | 55.9±1.2 | 90.6±0.9 | 77.6±1.4 | 77.1±1.2 | **85.5±1.0** | **82.4±1.2** | **82.0±1.3** |
| DART(kl src) | 78.9±1.0 | 55.5±1.0 | 54.6±1.6 | 92.0±0.1 | 78.1±0.8 | 77.5±0.6 | 84.6±0.3 | 81.0±0.5 | 80.5±0.6 |

Table 17: Standard accuracy (nat acc) / Robust accuracy under PGD attack (pgd acc)/ Robust accuracy under AutoAttack (aa acc) of target test data on PACS dataset with a fix source domain Clipart and different target domains.

| Source→Target | Art→Clipart | | | Art→Photo | | | Art→Sketch | | |
|---|---|---|---|---|---|---|---|---|---|
| Algorithm | clean acc | pgd acc | aa acc | clean acc | pgd acc | aa acc | clean acc | pgd acc | aa acc |
| Natural DANN | 84.3±0.6 | 12.4±6.1 | 1.1±0.8 | **97.9±0.4** | 2.7±1.5 | 0.0±0.0 | 84.9±0.7 | 0.6±0.3 | 17.5±5.2 |
| AT(src only) | 79.2±0.4 | 63.9±1.2 | 63.0±1.0 | 82.5±0.6 | 65.8±0.5 | 65.3±0.2 | 79.9±1.1 | 72.4±1.2 | 71.4±1.2 |
| TRADES(src only) | 81.5±1.5 | 62.5±2.0 | 60.8±1.9 | 87.9±0.6 | 70.8±0.8 | 69.9±0.6 | 78.8±1.1 | 71.5±0.7 | 70.7±0.6 |
| AT(tgt,pseudo) | 85.1±0.1 | 76.5±0.9 | 76.2±0.8 | 95.0±1.4 | **83.1±1.2** | **82.2±1.2** | 84.8±0.1 | 81.1±0.6 | 81.0±0.6 |
| TRADES(tgt,pseudo) | 85.1±0.9 | 77.2±1.0 | 76.9±1.0 | 95.3±0.7 | 82.2±0.6 | 81.4±0.4 | 86.1±0.7 | 83.3±0.7 | 83.0±0.7 |
| AT+UDA | 78.5±1.8 | 65.1±0.6 | 64.5±0.5 | 79.0±2.0 | 57.8±2.1 | 57.3±1.9 | 80.8±0.7 | 71.6±0.3 | 70.4±0.6 |
| ARTUDA | 88.3±2.1 | 76.0±1.7 | 74.3±2.1 | 95.0±0.6 | 78.5±1.3 | 74.7±1.3 | 80.3±1.3 | 61.5±1.0 | 53.5±1.8 |
| SROUDA | 84.2±1.4 | 75.8±0.6 | 75.1±0.5 | 94.1±0.7 | 81.5±1.0 | 80.3±1.1 | 77.3±4.6 | 73.2±4.9 | 72.6±4.8 |
| DART(clean src) | 89.1±0.3 | 79.1±0.3 | 78.7±0.2 | 95.9±0.7 | 81.4±1.4 | 80.3±1.6 | **89.5±0.6** | **86.4±0.5** | **85.8±0.6** |
| DART(adv src) | 88.9±0.4 | 79.2±0.9 | 78.3±1.0 | 94.1±0.4 | 81.3±0.6 | 80.6±0.8 | 87.9±0.9 | 84.6±0.9 | 84.3±0.9 |
| DART(kl src) | **89.4±0.7** | **80.9±1.0** | **80.5±1.2** | 96.1±0.5 | 81.1±0.4 | 80.5±0.4 | 88.3±0.6 | 85.4±0.5 | 85.1±0.5 |

Table 18: Standard accuracy (nat acc) / Robust accuracy under PGD attack (pgd acc)/ Robust accuracy under AutoAttack (aa acc) of target test data on PACS dataset with a fix source domain Art and different target domains.

| Source→Target | Sketch→Art | | | Sketch→Clipart | | | Sketch→Photo | | |
|---|---|---|---|---|---|---|---|---|---|
| Algorithm | clean acc | pgd acc | aa acc | clean acc | pgd acc | aa acc | clean acc | pgd acc | aa acc |
| Natural DANN | 68.0±1.2 | 1.5±1.2 | 0.7±0.5 | 72.3±1.4 | 14.0±4.5 | 7.4±3.1 | 71.1±4.0 | 0.3±0.1 | 0.0±0.0 |
| AT(src only) | 22.3±1.4 | 19.0±0.9 | 18.6±1.1 | 66.6±1.7 | 40.2±1.1 | 38.7±1.5 | 31.7±5.5 | 22.7±1.1 | 22.2±1.3 |
| TRADES(src only) | 26.8±4.2 | 17.3±2.1 | 11.6±4.8 | 67.1±0.8 | 43.4±1.3 | 42.6±1.4 | 35.0±5.5 | 21.3±2.4 | 12.3±4.3 |
| AT(tgt,pseudo) | 62.4±2.2 | 37.5±0.8 | 36.7±0.5 | 72.5±1.8 | 64.0±1.2 | 63.8±1.1 | 88.8±0.7 | 73.6±0.7 | 72.9±0.7 |
| TRADES(tgt,pseudo) | 69.1±2.3 | 44.2±2.5 | 43.2±2.2 | 71.6±0.6 | 63.8±0.6 | 63.4±0.4 | 89.6±0.7 | 76.4±1.4 | 75.6±1.4 |
| AT+UDA | 47.0±1.2 | 28.5±1.4 | 6.2±1.0 | 67.9±1.6 | 40.4±0.8 | 38.2±0.6 | 32.4±3.3 | 27.2±2.1 | 12.5±4.3 |
| ARTUDA | 49.5±2.4 | 31.7±3.4 | 31.1±3.2 | 38.1±2.5 | 25.5±1.8 | 22.9±1.4 | 48.9±1.8 | 40.4±2.9 | 39.6±3.2 |
| SROUDA | 24.5±1.7 | 22.4±0.3 | 22.4±0.4 | 72.4±1.0 | 62.3±0.2 | 61.3±0.2 | **91.9±0.5** | 73.1±3.6 | 70.5±4.7 |
| DART(clean src) | **71.9±1.8** | **53.1±4.4** | **52.4±4.6** | 78.4±0.7 | 69.2±0.9 | 68.9±0.8 | 87.8±1.4 | 76.8±1.0 | 75.9±1.1 |
| DART(adv src) | 67.8±1.4 | 47.4±2.9 | 46.6±3.0 | 77.3±0.8 | 68.2±1.0 | 67.9±1.1 | 89.3±0.8 | 77.6±1.5 | 77.0±1.7 |
| DART(kl src) | 69.4±0.5 | 49.3±1.4 | 48.6±1.6 | **80.0±0.5** | **70.3±0.2** | **70.1±0.2** | 90.5±0.5 | **77.7±1.7** | **77.2±1.9** |

Table 19: Standard accuracy (nat acc) / Robust accuracy under PGD attack (pgd acc)/ Robust accuracy under AutoAttack (aa acc) of target test data on PACS dataset with a fix source domain Sketch and different target domains.

## D.4  VISDA

| Source→Target | Synthetic→Real | | | Real→Synthetic | | |
|---|---|---|---|---|---|---|
| Algorithm | clean acc | pgd acc | aa acc | clean acc | pgd acc | aa acc |
| Natural DANN | 67.4±0.2 | 0.5±0.2 | 0.0±0.0 | 78.6±0.9 | 0.8±0.1 | 0.0±0.0 |
| AT(src only) | 19.0±0.2 | 18.0±0.3 | 17.2±0.4 | 53.5±0.8 | 41.6±0.4 | 39.8±0.4 |
| TRADES(src only) | 18.6±0.1 | 16.5±0.7 | 16.4±0.7 | 54.4±0.5 | 42.6±0.5 | 41.3±0.6 |
| AT(tgt, fix) | **69.6±0.3** | **58.3±0.7** | 57.5±0.7 | 85.7±0.2 | 82.0±0.2 | 81.7±0.2 |
| TRADES(tgt, fix) | 68.1±0.7 | 57.9±0.5 | 56.9±0.5 | 85.1±0.3 | 81.5±0.5 | 81.2±0.5 |
| AT+UDA | 48.0±1.1 | 24.1±0.9 | 18.5±1.4 | 66.4±0.6 | 66.4±0.6 | 47.8±0.8 |
| ARTUDA | 45.2±4.8 | 32.5±2.7 | 31.9±2.6 | 72.5±2.5 | 62.6±0.3 | 60.6±0.4 |
| SROUDA | 48.2±2.7 | 33.4±0.7 | 30.8±0.7 | 81.2±1.4 | 72.9±1.3 | 71.7±1.6 |
| DART(clean src) | 69.5±0.2 | 58.0±0.5 | 57.5±0.6 | **87.3±0.3** | **85.3±0.2** | 85.1±0.3 |
| DART(adv src) | 69.0±0.4 | 57.5±0.8 | 56.9±0.9 | 86.3±0.7 | 84.4±0.7 | 84.3±0.7 |
| DART(kl src) | 69.6±1.2 | 57.4±1.2 | **58.5±0.6** | 86.8±0.3 | 85.3±0.3 | **85.2±0.3** |

Table 20: Standard accuracy (nat acc) / Robust accuracy under PGD attack (pgd acc)/ Robust accuracy under AutoAttack (aa acc) of target test data on VisDA dataset.

# E  ADDITIONAL EXPERIMENTAL RESULTS

## E.1  RESULTS ON SOURCE TEST DATA

While our primary objective is to defend against attacks on the target domain, we note that DART continues to exhibit robustness against adversarial attacks on the source domain. In Table 21, we provide the standard and robust accuracy for the PGD attack on source test data. These results clearly demonstrate that DART, when employing an adversarial source or KL source, consistently maintains or even improves robustness on source test data.

| Dataset | DIGIT | | OfficeHome | | PACS | | VisDA | |
|---|---|---|---|---|---|---|---|---|
| Algorithm | nat acc | pgd acc | nat acc | pgd acc | nat acc | pgd acc | nat acc | pgd acc |
| Natural DANN | 96.5±0.1 | 85.7±0.1 | 71.2±0.2 | 1.2±0.1 | 89.7±0.4 | 12.3±1.4 | 88.0±0.7 | 1.9±0.2 |
| AT(src only) | 96.7±0.1 | 90.4±0.2 | 68.9±0.9 | 47.1±0.3 | 82.9±1.6 | 64.6±1.5 | 60.7±6.1 | 47.8±4.9 |
| TRADES(src only) | 96.5±0.0 | 90.5±0.2 | 68.8±0.2 | **47.5±0.4** | 81.5±2.2 | 62.5±1.9 | 72.2±1.8 | 57.7±1.6 |
| AT(tgt,pseudo) | 76.6±0.4 | 68.2±0.4 | 45.5±0.2 | 28.0±0.1 | 52.2±1.4 | 33.3±1.1 | 37.7±2.2 | 28.7±1.2 |
| TRADES(tgt,pseudo) | 79.3±1.2 | 69.6±0.7 | 45.2±0.8 | 29.6±0.3 | 55.2±1.1 | 36.7±0.7 | 37.9±0.6 | 28.9±0.5 |
| AT+UDA | 96.2±0.2 | 89.6±0.3 | 66.9±0.9 | 45.0±0.3 | 83.4±1.0 | 64.5±1.4 | 82.8±1.0 | 68.0±0.9 |
| ARTUDA | 96.1±0.1 | 82.8±0.0 | 67.3±0.3 | 38.4±0.2 | 85.1±1.0 | 47.0±0.2 | 81.0±3.1 | 31.5±2.1 |
| SROUDA | 82.6±0.2 | 73.0±0.3 | 46.2±0.2 | 30.4±0.2 | 58.3±1.4 | 33.7±1.5 | 35.1±4.0 | 19.7±1.6 |
| DART(clean src) | 96.0±0.0 | 82.4±0.2 | 65.1±0.7 | 37.6±0.4 | 85.1±0.1 | 51.5±0.6 | 80.9±2.0 | 38.6±2.0 |
| DART(adv src) | 96.1±0.0 | **90.5±0.1** | 65.0±0.6 | 45.7±0.4 | 86.0±0.8 | **68.7±0.5** | 77.8±0.8 | **64.3±1.1** |
| DART(kl src) | 95.9±0.0 | 89.4±0.1 | 64.9±0.4 | 44.6±0.3 | 85.5±0.1 | 67.4±0.3 | 80.3±1.2 | 62.8±1.2 |

Table 21: Standard / Robust accuracy(%) of source test data with an average of all source-target pairs for all datasets. These experiments compare 11 algorithms across 46 source-target pairs in the exact same conditions.

## E.2  ROBUSTNESS FOR VARYING ATTACK ITERATIONS

In Figure 2, we present a plot of the robust accuracy vs. the number of attack iterations (for a fixed perturbation size of $2/255$). The results indicate that 20 attack iterations are sufficient to achieve the strongest PGD attack within the given perturbation size.

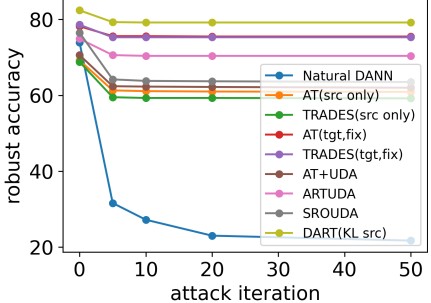

Figure 2: Robust accuracy as a function of attack iterations for different algorithms on PACS ( Photo→Sketch).

