# OpenReview forum: "DART: A Principled Approach to Adversarially Robust Unsupervised Domain Adaptation"
_ICLR.cc/2024/Conference — Submitted to ICLR 2024_

### Official Review · Reviewer_vW45 · 2023-10-26

**Soundness:** 1 poor
**Presentation:** 2 fair
**Contribution:** 2 fair
**Rating:** 3
**Confidence:** 4

**Summary:**

This work aims to improve the adversarial robustness of UDA methods. The authors first provide an upper bound for the adversarial target error, showing that the target adversarial error can be bounded by the source error and the discrepancy. They further approximate the ideal joint loss and propose a practical bound. The algorithm is then implemented based on the error bound. Some experiments show that the proposed method achieves better adversarial robustness than baselines.

**Strengths:**

Considering the target domain lacks labeled data, it is indeed difficult to gain robustness from other domains with domain shift. So, I admit that the authors focus on a challenging and interesting problem.

Besides, the authors implement a testbed for evaluating the robustness under the UDA setting, which is a vital contribution to this field.

**Weaknesses:**

After reading this paper, I have some concerns:
After reading this paper, I have some concerns:
* My primary concern is that there may be a gap between the theory and the algorithm. Specifically, in order to upper bound and optimize the ideal joint loss, the authors use one of the hypothesis $h\in \mathcal{H}$ to replace the optimal hypothesis $h^*$. I don't think this is appropriate. By the definition of the ideal joint loss, it is the minimal value optimized on $\mathcal{H}$. Hence, given a fixed hypothesis class, this term is fixed. Let's denote the ideal joint loss and its upper bound as $\gamma$ and $\gamma^{+}$ respectively. In the algorithm, the learner aims to minimize the $\gamma^+$. And they explain in Section 3 that it is beneficial to control this term. However, in my opinion, this ideal joint loss is a fixed value and cannot be optimized by minimizing its upper bound. The optimization of this term seems somewhat ridiculous. Although the authors claim in the experiments that optimizing this term is empirically beneficial for robustness, I am inclined to believe that the minimization of the target adversarial error is a heuristic as in prior works, which disproves the authors' claim that their algorithm is theoretically justified.
* In addition, the algorithm needs to generate pseudo-labels on the target domain, while the theoretical bound involves the adversarial target error, which needs the ground truth labels. The authors don't take the effect of incorrect pseudo-labels into consideration. This is another gap between the theory and the practical algorithm.
* The optimization process of Eq. (6) in the pseudocode of Algorithm 1 is not clear. In fact, the optimization is complex, involving many steps such as the generation of adversarial examples and the optimization of $f,g$.  It would be better to provide a more clear explanation of the algorithm.

**Questions:**

Why does the algorithm achieve the best clean accuracy compared with the baselines in DIGIT datasets? The proposed algorithm involves adversarial training, which is assumed to hurt clean accuracy.

---

> ### Author Response · Authors · 2023-11-15
> **Response to Reviewer vW45 [1/2]**
>
> Thank you for the comments and suggestions.
>
> [W1]
> > “However, in my opinion, this ideal joint loss is a fixed value and cannot be optimized by minimizing its upper bound. The optimization of this term seems somewhat ridiculous.”
>
> Thank you for the discussion. We’d like to point out that the ideal joint risk is not a fixed quantity; instead, it’s a function of the feature representations (that we are learning), so controlling it makes sense. This is not a claim that we are making in this paper, but a mathematical fact, which is clearly stated in the seminal work [1]. To simplify the discussion, we will first discuss why this holds using [1]’s notation and setup, and after that we will discuss why it holds in our setup (which is the same as [1] but with adversarial robustness).
>
> In Ben-David et al. (2006) [1], the authors fix a feature representation function R, and consider a hypothesis class H that takes feature representations (based on R) as input–so the hypothesis class H is a function of R. The ideal joint risk is by definition a function of H (for a given/fixed R). Theorem 2 in [1] is stated for a fixed R and H, and motivated by the theorem, the authors in [1] and also in DANN’s paper [2] call for finding/learning an R that minimizes the upper bound in Theorem 2. Thus, every time the representation function R is changed (e.g., by a learning algorithm as in DANN), the hypothesis class H will change (because it is a function of R), and consequently the ideal joint loss will change (because it is a function of H). Now in [1] the authors chose to *assume* that the ideal joint loss is small for any “reasonable R”; quoting from the authors “we assume that $\lambda$ is small for reasonable R”. Of course there could be cases where the chosen R will lead to a large ideal joint loss, so controlling this loss will be important in such cases.
>
> In our setup, the ideal joint loss is defined similarly to [1] with the exception that the target examples are transformed by an adversary (as described in eq (2) in our paper). Similar to [1], our ideal joint loss is a function of the feature representation. However, please note that we are using a different notation than [1]. In our notation H represents the hypothesis class of $h:= f \circ g$, i.e., H is the hypothesis class of the composition of the feature representation function g (called R in [1]) and the main classifier f. Formally, $f\in F, g\in G, H=\\{h:h=f \circ g, s.t. f\in F, g\in G\\}$. Thus, every time we change the feature representation g, the ideal joint loss is optimized over a subset of H, i.e, given the fixed g, we optimize h over $H(g)=\\{h:h=f \circ g, s.t. f\in F\\}$, and consequently the ideal joint loss will change. Thus, generally, the learning algorithm may encounter g’s that lead to a large ideal joint risk–by attempting to control this term (even loosely) we may be able to avoid such bad g’s.
>
> In the revised version of the paper, we added a clarification on this point -- please see the end of Section 3 and in the paragraph “A practical bound” in Section 4.
>
> [1] Ben-David, S., Blitzer, J., Crammer, K., & Pereira, F. (2006). Analysis of representations for domain adaptation. Advances in neural information processing systems, 19.
>
> [2] Ganin, Yaroslav, Evgeniya Ustinova, Hana Ajakan, Pascal Germain, Hugo Larochelle, François Laviolette, Mario Marchand, and Victor Lempitsky. "Domain-adversarial training of neural networks." The journal of machine learning research 17, no. 1 (2016): 2096-2030.
>
> [W2] Please note that we are not claiming that we directly control the ideal joint risk in the practice and we agree that there’s a gap between the algorithm and theory; basically the algorithm “attempts” to control a proxy of the ideal joint loss (using pseudo labels). Whether this control is effective or not is ultimately decided by the empirical results, and our experiments show that controlling this proxy is helpful on many datasets–please see the ablation study in Table 2.
>
> As a side note, all of the major practical works on UDA (e.g., DANN) are motivated by theory but also have a “gap” between theory in practice. E.g., in DANN, the theory calls for evaluating H-divergence which requires optimizing a 0/1 loss to global optimality–this is impossible in practice so DANN just uses a simple heuristic algorithm (SGD and logistic loss) to approximate H divergence. Moreover, in DANN the ideal joint risk (which is a function of the feature representation) is completely ignored–another gap from the theory.
>
> However, we believe that such gaps (whether in our work or others such as DANN) should not be a reason for rejection, especially given the strong empirical results. Moreover, the theory in our case helped uncover key quantities that should be controlled in the adversarial robustness UDA setup–these quantities were not thoroughly studied in previous work on the same problem.

---

> > ### Author Response · Authors · 2023-11-15
> > **Response to Reviewer vW45 [2/2]**
> >
> > [W3] At each iteration, we first fix f and g to generate the adversarial examples via projected gradient descent (PGD). We then fix the adversarial examples and update the model f,g via stochastic gradient descent (SGD). The procedure follows the same routines as adversarial training [Madry et al 2018]. We further provide a detailed discussion by using DART with DANN as shown in Section B.1.
> >
> > [Q] While we agree that under standard ERM adversarial training typically (but not necessarily) reduces clean accuracy, the same observation may not be as typical in transfer learning. We hypothesize DART achieves the highest clean accuracy in DIGIT datasets for two reasons.
> >
> > First, using pseudo labels of the target examples may help in domain adaptation. On DIGITs, all algorithms that use pseudo labels seem to significantly outperform standard DANN in terms of clean accuracy. In our ablation study, we compare DART w/o third term (i.e., version of DART that doesn’t incorporate pseudo labels) with DART with all components (see Table 2 in the paper), and DART with all components achieves higher clean performance compared with DART w/o third term. In the table below we also report the clean accuracy of natural DANN, DART (with all components), and DART w/o third term. The results indicate that removing pseudo labels makes DART more comparable to natural DANN.
> >
> >
> > Table: Natural (clean) accuracy. “DART w.o. Third term” does not use pseudo labels.
> > |    |PACS Photo $\to$ Sketch      | SYN$\to$MNIST-M | SVHN $\to$MNIST|
> > | ----------- | ----------- |----------- | ----------- |
> > | Natural DANN      | 74.0 ± 1.1       | 60.4 ± 0.8 | 82.6 ± 0.2 |
> > | DART   | 82.5 ± 0.8        | 75.2 ± 0.8 | 98.2 ± 0.2 |
> > | DART w.o. third term| 72.3 ± 2.4 | 65.7 ± 0.5 | 84.8 ± 0.3 |
> >
> >
> >
> > We also note that this aligns with the work of [1] (SroUDA), where adversarial training while using pseudo labels improves clean accuracy on some datasets (e.g., see Fig 1 in [1]).
> >
> > Second, adversarial-training style algorithms may help improve clean performance in the presence of distribution shifts. There are a series of works (e.g., [2,3,4]) that have studied such phenomena both theoretically and empirically.
> >
> > [1] Zhu, Wanqing, et al. "SRoUDA: meta self-training for robust unsupervised domain adaptation." Proceedings of the AAAI Conference on Artificial Intelligence. Vol. 37. No. 3. 2023.
> >
> > [2] Shafahi, Ali, et al. "Adversarially robust transfer learning." Eighth International Conference on Learning Representations, 2020.
> >
> > [3] Salman, Hadi, et al. "Do adversarially robust imagenet models transfer better?." Advances in Neural Information Processing Systems 33, 2020.
> >
> > [4] Deng, Zhun, et al. "Adversarial training helps transfer learning via better representations." Advances in Neural Information Processing Systems 34, 2021.

---

> > > ### Author Response · Authors · 2023-11-21
> > > **Follow-up on rebuttal**
> > >
> > > Dear Reviewer,
> > >
> > > Thank you again for your valuable comments! We have carefully addressed your comments.
> > >
> > > Please let us know if our replies have satisfactorily addressed your concerns. Since the discussion period will be closed soon, please do not hesitate to let us know if you have any further questions or if you require any additional clarification. Thank you in advance!

---

### Official Review · Reviewer_d4Ff · 2023-10-27

**Soundness:** 2 fair
**Presentation:** 3 good
**Contribution:** 2 fair
**Rating:** 6
**Confidence:** 3

**Summary:**

In this work, the authors provided an efficient algorithm DART to train a model which is adversarially robust in the target domain. Their method is motivated by an adversarial error bound and does not need to change the framework of DANN. The experiments validate the effectiveness of their method.

**Strengths:**

1. This paper is well-written.
2. The adversarial robustness of the model on target domain is largely improved by the proposed method. And DART does not compromise the standard accuracy. This improvement represents a significant advancement in this research field.
3. The algorithm can be generally applied based on various prior methods. We only need to change a few codes based on the original frameworks.

**Weaknesses:**

I am mainly concerned about the experiments:
1. More expeiments on various neural networks are encouraged. Will the algorithm still perform well on smaller networks such as ResNet-18, or larger networks such as WideResNet?
2. More expeiments on Office-31 are encouraged, since this is also an important dataset widely used in UDA setting.
3. The adversarial perturbation $2/255$ is small. What is the performance of DART under larger perturbation, such as $8/255$?

If the authors can address my questions, I would reconsider the rate.

**Questions:**

Please refer to the Weaknesses.

---

> ### Author Response · Authors · 2023-11-15
> **Response to Reviewer d4Ff**
>
> Thank you for the comments and suggestions.
>
> [W1] Please note that we have tried two different network architectures, i.e., small CNN for DIGIT datasets, and ResNet50 for PACS, OfficeHome, and VisDA. These are the standard popular network architectures that have been used in the transfer learning literature [1,2,3,4]. The results are consistent across the two architectures.
>
> [W2] We have included a variety of datasets (with experiments covering 46 source-target pairs), and all of these datasets are much larger and arguably more complex than Office31 (e.g., OfficeHome, PACS, VisDA). Having said that, we're currently running an additional experiment with Office31 and will report the results back if the experiment finishes before the rebuttal deadline.
>
>
> [W3] For the chosen models and datasets, we think that eps=2/255 is reasonable choice for a few reasons:
> - eps=2/255 significantly degrades the performance of natural DANN (without any defense mechanism), e.g., the robust accuracy is lower than 2% for OfficeHome and VisDA. So this eps is non-trivial and is reasonable to use (from an attacker’s perspective) because it leads to a relatively low image distortion while significantly fooling the model.
> - In figure 1, we present the results for models trained with eps=2/255 but tested with a different eps varying between 1/255 and 64/255. The results indicate that DART consistently outperforms other algorithms across the whole range of testing epsilons.
> - Also please note that larger epsilons may generally require more iterations for the attack to converge (whether during training or inference), assuming a fixed learning rate for the attack. Since we’re carrying out large-scale experiments (training more than 29700 models), this can significantly increase the compute time.
>
> As a side note, there are many works in the literature where a similar epsilon is used, e.g.,  [1,2,3] also trained or evaluated with 2/255 l_infty perturbation size.
>
> With that said, based on your suggestion, we have started running a set of experiments with epsilon=8/255 and will report the result once it’s finished.
>
>
> [1] Gulrajani, Ishaan, and David Lopez-Paz. "In search of lost domain generalization." The Ninth International Conference on Learning Representations (2020).
>
> [2] Ye, Haotian, et al. "Towards a theoretical framework of out-of-distribution generalization." Advances in Neural Information Processing Systems 34 (2021).
>
> [3] Rame, Alexandre, et al. "Diverse weight averaging for out-of-distribution generalization." Advances in Neural Information Processing Systems 35 (2022).
>
> [4] Chu, Xu, et al. "DNA: Domain generalization with diversified neural averaging." International Conference on Machine Learning. PMLR, 2022.
>
> [5] Balunovic, Mislav, and Martin Vechev. "Adversarial training and provable defenses: Bridging the gap." International Conference on Learning Representations. 2019.
>
> [6] Sehwag, Vikash, et al. "Hydra: Pruning adversarially robust neural networks." Advances in Neural Information Processing Systems 33 (2020).
>
> [7] Tao, Lue, et al. "Can Adversarial Training Be Manipulated By Non-Robust Features?." Advances in Neural Information Processing Systems 35 (2022).

---

> > ### Author Response · Authors · 2023-11-18
> > **Additional experiment result on perturbation size 8/255**
> >
> > We conducted additional experiments with epsilon=8/255 on four DIGIT source-target pairs (fixing one source and trying all possible targets), comparing DART with some of the most competitive methods. The algorithms were trained and evaluated using the same epsilon.  The results, as shown in the tables below, indicate that DART outperforms other methods on average. These findings are consistent with the previous results in the paper. We will provide the full results in the final version. Thanks again for this suggestion.
> >
> >
> > | Algorithm      | SVHN -> MNIST | SVHN -> MNIST-M | SVHN->SYN | SVHN->USPS |
> > | ----------- | ----------- | ----------- | ----------- | ----------- |
> > | Algorithm      | nat acc$~~~~~~~~~$pgd acc | nat acc$~~~~~~~~~$pgd acc | nat acc$~~~~~~~~~$pgd acc | nat acc$~~~~~~~~~$pgd acc | nat acc$~~~~~~~~~$pgd acc |
> > | AT (tgt)      |   88.0 $\pm$ 0.1, 84.5 $\pm$ 0.1    | 62.5 $\pm$ 0.9, 43.2 $\pm$ 0.3  |  95.8 $\pm$ 0.0, 87.2 $\pm$ 0.1 |  95.5 $\pm$ 0.4, 90.5 $\pm$ 0.3
> > | TRADES (tgt)   |   88.2 $\pm$ 0.4, 84.9 $\pm$ 0.6   |  **63.1 $\pm$ 1.5**, 41.5 $\pm$ 1.4  |   96.0 $\pm$ 0.1, 89.1 $\pm$ 0.4 |  95.5 $\pm$ 0.2, 91.4 $\pm$ 0.4
> > | ARTUDA   |    96.8 $\pm$ 0.7, 94.2 $\pm$ 1.2    |  48.6 $\pm$ 2.5, 24.1 $\pm$ 2.6    |  95.2 $\pm$ 0.2, 82.7 $\pm$ 0.4 |  98.5 $\pm$ 0.1, 94.0 $\pm$ 0.8
> > | SROUDA   |   87.0 $\pm$ 0.3, 80.4 $\pm$ 0.4  |  53.6 $\pm$ 1.7, 39.6 $\pm$ 1.1   | 96.4 $\pm$ 0.1, **89.1 $\pm$ 0.1** |  96.5 $\pm$ 0.2, 88.8 $\pm$ 1.0
> > | DART   |     **99.0 $\pm$ 0.0**, **97.8 $\pm$ 0.2**    |  62.1 $\pm$ 0.4, **45.9 $\pm$ 0.1**   | **97.1 $\pm$ 0.1**, 88.6 $\pm$ 0.4  |  **98.8 $\pm$ 0.1**, **96.2 $\pm$ 0.1**
> >
> > *Table: Standard accuracy (nat acc) / Robust accuracy under PGD attack (pgd acc) of target test data on DIGIT dataset with a fix source domain SVHN and different target domains.*

---

> > > ### Author Response · Authors · 2023-11-21
> > > **More experiment results on perturbation size 8/255**
> > >
> > > Here we provide more results with epsilon=8/255 on three PACS source-target pairs (fixing one source and trying all possible targets). The results, as shown in the tables below, indicate that DART outperforms other methods on average in terms of robust accuracy. These findings are consistent with the previous results in the paper.
> > >
> > > | Source$\to$Target       | Photo -> Sketch | Photo ->Clipart | Photo->Art |
> > > | ----------- | ----------- | ----------- | ----------- |
> > > | Algorithm      | nat acc$~~~~~~~~~$pgd acc | nat acc$~~~~~~~~~$pgd acc | nat acc$~~~~~~~~~$pgd acc | nat acc$~~~~~~~~~$pgd acc |
> > > | AT (tgt)      |   77.0 $\pm$ 0.5, 72.0 $\pm$ 0.5    | 81.9 $\pm$ 0.6, 65.1 $\pm$ 1.5  |  33.8 $\pm$ 1.2, 26.9 $\pm$ 0.5
> > > | TRADES (tgt)   |   77.2 $\pm$ 0.5, 72.5 $\pm$ 0.6   |  80.9 $\pm$ 1.2, 66.5 $\pm$ 1.3  |   62.6 $\pm$ 3.4, **29.7 $\pm$ 1.0**
> > > | ARTUDA   |    63.4 $\pm$ 4.9, 57.1 $\pm$ 3.5    |  71.1 $\pm$ 2.8, 52.9 $\pm$ 1.7    |  26.6 $\pm$ 0.7, 23.5 $\pm$ 1.0
> > > | SROUDA   |   50.2 $\pm$ 3.8, 41.9 $\pm$ 2.7 |  81.2 $\pm$ 1.4, 64.1 $\pm$ 0.7   | 29.0 $\pm$ 0.6, 24.4 $\pm$ 0.8
> > > | DART   |     79.9 $\pm$ 0.9, **74.0 $\pm$ 0.9**   |  85.3 $\pm$ 0.8, **68.9 $\pm$ 1.6**   | 52.8 $\pm$ 6.0, 28.1 $\pm$ 2.6
> > >
> > > *Table: Standard accuracy (nat acc) / Robust accuracy under PGD attack (pgd acc) of target test data on PACS dataset with a fix source domain Photo and different target domains.*

---

> > > > ### Author Response · Authors · 2023-11-21
> > > > **Additional experiment result on Office31**
> > > >
> > > > Here we provide more results with epsilon=8/255 on Office31 as suggested by the reviewer. We compare with the SOTA of adversarial robustness UDA baselines. The results, as shown in the table below, indicate that DART outperforms the existing competitive baselines.
> > > >
> > > > | Source$\to$Target       | Amazon -> Webcam | Webcam ->Amazon |
> > > > | ----------- | ----------- | ----------- |
> > > > | Algorithm      | nat acc$~~~~~~~~~$pgd acc | nat acc$~~~~~~~~~$pgd acc | nat acc$~~~~~~~~~$pgd acc |
> > > > | ARTUDA   |    52.6 $\pm$ 8.3, 36.7 $\pm$ 7.2    |  41.0 $\pm$ 3.9, 32.6 $\pm$ 2.3    |
> > > > | SROUDA   |   61.6 $\pm$ 1.9, 47.2 $\pm$ 2.8 |  55.6 $\pm$ 0.9, 45.1 $\pm$ 1.0   |
> > > > | DART   |     **80.7 $\pm$ 1.2**, **55.6 $\pm$ 1.6**   |  **58.1 $\pm$ 1.2**, **47.2 $\pm$ 0.8**   |
> > > >
> > > > | Source$\to$Target       | DSLR -> Amazon | Amazon ->DSLR |
> > > > | ----------- | ----------- | ----------- |
> > > > | Algorithm      | nat acc$~~~~~~~~~$pgd acc | nat acc$~~~~~~~~~$pgd acc | nat acc$~~~~~~~~~$pgd acc |
> > > > | ARTUDA   |    32.3 $\pm$ 3.1, 26.3 $\pm$ 2.0    |  33.0 $\pm$ 1.7, 13.3 $\pm$ 1.4   |
> > > > | SROUDA   |   55.5 $\pm$ 0.9, 43.3 $\pm$ 0.9 |  47.0 $\pm$ 3.4, 28.3 $\pm$ 1.7   |
> > > > | DART   |     **61.6 $\pm$ 0.9**, **44.9 $\pm$ 2.3**   |  **69.3 $\pm$ 2.4**, **35.0 $\pm$ 3.7**   |
> > > >
> > > > | Source$\to$Target       | DSLR -> Webcam | Webcam ->DSLR |
> > > > | ----------- | ----------- | ----------- |
> > > > | Algorithm      | nat acc$~~~~~~~~~$pgd acc | nat acc$~~~~~~~~~$pgd acc | nat acc$~~~~~~~~~$pgd acc |
> > > > | ARTUDA   |    78.6 $\pm$ 3.6, 57.7 $\pm$ 2.8    |  81.7 $\pm$ 5.2, 56.3 $\pm$ 3.1   |
> > > > | SROUDA   |   88.1 $\pm$ 1.7, 70.0 $\pm$ 2.7 |  74.3 $\pm$ 3.1, 50.3 $\pm$ 3.3   |
> > > > | DART   |     **92.9 $\pm$ 0.6**, **75.5 $\pm$ 2.1**   |  **93.3 $\pm$ 0.5**, **61.7 $\pm$ 3.0**   |
> > > >
> > > > *Table: Standard accuracy (nat acc) / Robust accuracy under PGD attack (pgd acc) of target test data on Office31 dataset.*

---

> > > > > ### Author Response · Authors · 2023-11-21
> > > > > **Follow-up on rebuttal**
> > > > >
> > > > > Dear Reviewer,
> > > > >
> > > > > Thank you again for your valuable comments! We have carefully addressed your comments.
> > > > >
> > > > > Please let us know if our replies have satisfactorily addressed your concerns. Since the discussion period will be closed soon, please do not hesitate to let us know if you have any further questions or if you require any additional clarification. Thank you in advance!

---

### Official Review · Reviewer_qNjr · 2023-10-28

**Soundness:** 3 good
**Presentation:** 3 good
**Contribution:** 3 good
**Rating:** 6
**Confidence:** 3

**Summary:**

In this work, the authors study the problem of adversarial robustness under distribution shift, i.e., UDA. In order to overcome this problem, they first provide a generalization bound for the adversarial target loss, consisting three terms that can be optimized. Hence, they present a theoretically guaranteed algorithm to improve the adversarial robustness on the target domain. DART significantly improves the robustness on various adaptation tasks.

**Strengths:**

+ Originality: There are some related works that also try to improve adversarial robustness for UDA setting. However, this work is different from the prior works.

+ Quality: This work is theoretically motivated and is more empirically efficient than the baselines.

+ Clarity: The writing is clear generally.

+ Significance: The adversarial robustness is essential in machine learning, especially for the safety-critical applications.

**Weaknesses:**

* The theoretical results lack novelty. It seems that Theory 3.1 is similar to the bound in [1]. The authors only need to replace the standard loss function with the adversarial loss function.
* The perturbation radius is usually set to $\frac{8}{255}$ or $\frac{16}{255}$. In this work, the authors set the radius to $\frac{2}{255}$. This may not be sufficient to illustrate the efficacy of the proposed method. I noticed that the authors conducted experiments with various radii on the task Photo -> Sketch. However, I think more experiments on other datasets are needed.

[1] A theory of learning from different domains. Shai Ben-David, et al.

**Questions:**

* In Table 3, the authors compare the proposed algorthm with AT (tgt,cg). What is the difference between these two methods? Why does DART have better robustness compared to AT (tgt,cg) ?

**Details Of Ethics Concerns:**

I have no ethical concerns.

---

> ### Author Response · Authors · 2023-11-15
> **Response to Reviewer qNjr**
>
> Thank you for the comments and suggestions.
>
> [W1]
> > “The theoretical results lack novelty. It seems that Theory 3.1 is similar to the bound in [1]. The authors only need to replace the standard loss function with the adversarial loss function.”
>
> Our result is inspired by [1] but does not directly follow from [1]. The proof of Theorem 2 in [1] does not follow if we replace the standard loss with adversarial loss. Specifically, the proof technique in [1] is not sufficient to obtain a generalization bound (from sample to population) for the adversarial version of H divergence. To deal with this, we had to carefully decompose the adversarial version of H divergence (among other terms in the objective).
>
>
> [W2] *Regarding perturbation size.* For the chosen models and datasets, we think that eps=2/255 is reasonable choice for a few reasons:
> - eps=2/255 significantly degrades the performance of natural DANN (without any defense mechanism), e.g., the robust accuracy is lower than 2% for OfficeHome and VisDA. So this eps is non-trivial and is reasonable to use (from an attacker’s perspective) because it leads to a relatively low image distortion while significantly fooling the model.
> - In figure 1, we present the results for models trained with eps=2/255 but tested with a different eps varying between 1/255 and 64/255. The results indicate that DART consistently outperforms other algorithms across the whole range of testing epsilons.
> - Also please note that larger epsilons may generally require more iterations for the attack to converge (whether during training or inference), assuming a fixed learning rate for the attack. Since we’re carrying out large-scale experiments (training more than 29700 models), this can significantly increase the compute time.
>
> As a side note, there are many works in the literature where a similar epsilon is used, e.g.,  [1,2,3] also trained or evaluated with 2/255 $l_\infty$ perturbation size.
>
> With that said, based on your suggestion, we have started running a set of experiments with epsilon=8/255 and will report the result once it’s finished.
>
> [Q] AT (tgt,cg) is defined as applying adversarial training on the target examples with pseudo-labels, where the pseudo-labels are generated using exactly the same algorithm we use in DART (as described in Section 4 paragraph Pseudo-Labels $\hat Y_T$). Essentially, AT(tgt, cg) only considers the third term in our optimization formulation (equation 6). We hypothesize that DART outperforms AT(tgt,cg) because DART leverages additional terms (adversarial H-divergence and source loss).
>
>
> [1] Balunovic, Mislav, and Martin Vechev. "Adversarial training and provable defenses: Bridging the gap." International Conference on Learning Representations. 2019.
>
> [2] Sehwag, Vikash, et al. "Hydra: Pruning adversarially robust neural networks." Advances in Neural Information Processing Systems 33 (2020).
>
> [3] Tao, Lue, et al. "Can Adversarial Training Be Manipulated By Non-Robust Features?." Advances in Neural Information Processing Systems 35 (2022).

---

> ### Author Response · Authors · 2023-11-18
> **Additional experiment result on perturbation size 8/255**
>
> We conducted additional experiments with epsilon=8/255 on four DIGIT source-target pairs (fixing one source and trying all possible targets), comparing DART with some of the most competitive methods. The algorithms were trained and evaluated using the same epsilon.  The results, as shown in the tables below, indicate that DART outperforms other methods on average. These findings are consistent with the previous results in the paper. We will provide the full results in the final version. Thanks again for this suggestion.
>
>
> | Source$\to$Target      | SVHN -> MNIST | SVHN -> MNIST-M | SVHN->SYN | SVHN->USPS |
> | ----------- | ----------- | ----------- | ----------- | ----------- |
> | Algorithm      | nat acc$~~~~~~~~~$pgd acc | nat acc$~~~~~~~~~$pgd acc | nat acc$~~~~~~~~~$pgd acc | nat acc$~~~~~~~~~$pgd acc | nat acc$~~~~~~~~~$pgd acc |
> | AT (tgt)      |   88.0 $\pm$ 0.1, 84.5 $\pm$ 0.1    | 62.5 $\pm$ 0.9, 43.2 $\pm$ 0.3  |  95.8 $\pm$ 0.0, 87.2 $\pm$ 0.1 |  95.5 $\pm$ 0.4, 90.5 $\pm$ 0.3
> | TRADES (tgt)   |   88.2 $\pm$ 0.4, 84.9 $\pm$ 0.6   |  **63.1 $\pm$ 1.5**, 41.5 $\pm$ 1.4  |   96.0 $\pm$ 0.1, 89.1 $\pm$ 0.4 |  95.5 $\pm$ 0.2, 91.4 $\pm$ 0.4
> | ARTUDA   |    96.8 $\pm$ 0.7, 94.2 $\pm$ 1.2    |  48.6 $\pm$ 2.5, 24.1 $\pm$ 2.6    |  95.2 $\pm$ 0.2, 82.7 $\pm$ 0.4 |  98.5 $\pm$ 0.1, 94.0 $\pm$ 0.8
> | SROUDA   |   87.0 $\pm$ 0.3, 80.4 $\pm$ 0.4  |  53.6 $\pm$ 1.7, 39.6 $\pm$ 1.1   | 96.4 $\pm$ 0.1, **89.1 $\pm$ 0.1** |  96.5 $\pm$ 0.2, 88.8 $\pm$ 1.0
> | DART   |     **99.0 $\pm$ 0.0**, **97.8 $\pm$ 0.2**    |  62.1 $\pm$ 0.4, **45.9 $\pm$ 0.1**   | **97.1 $\pm$ 0.1**, 88.6 $\pm$ 0.4  |  **98.8 $\pm$ 0.1**, **96.2 $\pm$ 0.1**
>
> *Table: Standard accuracy (nat acc) / Robust accuracy under PGD attack (pgd acc) of target test data on DIGIT dataset with a fix source domain SVHN and different target domains.*

---

> ### Author Response · Authors · 2023-11-21
> **More experiment results on perturbation size 8/255**
>
> Here we provide more results with epsilon=8/255 on three PACS source-target pairs (fixing one source and trying all possible targets). The results, as shown in the tables below, indicate that DART outperforms other methods on average in terms of robust accuracy. These findings are consistent with the previous results in the paper.
>
> | Source$\to$Target      | Photo -> Sketch | Photo ->Clipart | Photo->Art |
> | ----------- | ----------- | ----------- | ----------- |
> | Algorithm      | nat acc$~~~~~~~~~$pgd acc | nat acc$~~~~~~~~~$pgd acc | nat acc$~~~~~~~~~$pgd acc | nat acc$~~~~~~~~~$pgd acc |
> | AT (tgt)      |   77.0 $\pm$ 0.5, 72.0 $\pm$ 0.5    | 81.9 $\pm$ 0.6, 65.1 $\pm$ 1.5  |  33.8 $\pm$ 1.2, 26.9 $\pm$ 0.5
> | TRADES (tgt)   |   77.2 $\pm$ 0.5, 72.5 $\pm$ 0.6   |  80.9 $\pm$ 1.2, 66.5 $\pm$ 1.3  |   62.6 $\pm$ 3.4, **29.7 $\pm$ 1.0**
> | ARTUDA   |    63.4 $\pm$ 4.9, 57.1 $\pm$ 3.5    |  71.1 $\pm$ 2.8, 52.9 $\pm$ 1.7    |  26.6 $\pm$ 0.7, 23.5 $\pm$ 1.0
> | SROUDA   |   50.2 $\pm$ 3.8, 41.9 $\pm$ 2.7 |  81.2 $\pm$ 1.4, 64.1 $\pm$ 0.7   | 29.0 $\pm$ 0.6, 24.4 $\pm$ 0.8
> | DART   |     79.9 $\pm$ 0.9, **74.0 $\pm$ 0.9**   |  85.3 $\pm$ 0.8, **68.9 $\pm$ 1.6**   | 52.8 $\pm$ 6.0, 28.1 $\pm$ 2.6
>
> *Table: Standard accuracy (nat acc) / Robust accuracy under PGD attack (pgd acc) of target test data on PACS dataset with a fix source domain Photo and different target domains.*

---

> > ### Author Response · Authors · 2023-11-21
> > **Follow-up on rebuttal**
> >
> > Dear Reviewer,
> >
> > Thank you again for your valuable comments! We have carefully addressed your comments.
> >
> > Please let us know if our replies have satisfactorily addressed your concerns. Since the discussion period will be closed soon, please do not hesitate to let us know if you have any further questions or if you require any additional clarification. Thank you in advance!

---

### Official Review · Reviewer_ag53 · 2023-10-30

**Soundness:** 2 fair
**Presentation:** 2 fair
**Contribution:** 1 poor
**Rating:** 3
**Confidence:** 5

**Summary:**

This paper develops a generalization bound for adversarial target error by incorporating the source loss, worst-case divergence, and the ideal joint loss. This leads to the introduction of a DANN-inspired algorithm aimed at enhancing robustness within the target domain. To validate both the theoretical framework and the effectiveness of the algorithm, a comprehensive set of empirical experiments is conducted.

**Strengths:**

This paper provides a novel error bound for the adversarial robustness of unsupervised domain adaptation. Compared to prior works that are heuristic, DART has theoretical support and higher performance.

**Weaknesses:**

From a theoretical perspective, the algorithm is more interesting than those heuristic methods. However, when I read the proofs of Theorem 3.1 in Appendix, I find that there are some mistakes which may affect the validity of the theory. At the bottom of page 14, the second equation in the proof is weird. By the definition of $\mathcal{P}(\mathcal{D}_T)$, $\tilde{\mathcal{D}}_T$ is the distribution of the adversarial examples. But in the third line, it is $(x,y)\sim \tilde{\mathcal{D}}_T$ rather than $(\tilde{x},y)\sim \tilde{\mathcal{D}}_T$. Actually I cannot find a suitable way to correct this mistake which prevent me reading the rest of the proof.

Besides, the significance of the proposed testbed DomainRobust is limited. As the authors state, DomainRobust is implemented based on the exist framework DomainBed. As far as I can see, they only need to add the implementation of the adversarial part, such as the PGD attack methods.

**Questions:**

None

---

> ### Author Response · Authors · 2023-11-15
> **Response to Reviewer ag53**
>
> [W1] Thank you for pointing out this typo in the proof. We believe that this is a simple typo: basically all $(\tilde x,y)$ should be $(x,y)$ starting from the second equation. We fixed this in the revised paper.  To clarify, starting the second equation, we consider $(x,y)\sim \tilde D_T$, where $\tilde D_T$ is the worst case (adversarial) target distribution, and all the equation only works on $(x,y)$ instead of $(\tilde x,y)$. The proof follows automatically.
>
> [W2] Regarding the comment on DomainRobust, we would like to emphasize that the contribution of DomainRobust extends beyond the mere addition of an adversarial component to DomainBed. We did a major redesign of DomainBed that required rewriting the algorithm code and major parts of eval from scratch. We added **~6000 lines of new code** on top of DomainBed. We clarify our contributions in more detail below:
>
> 1. Transfer learning setting: DomainBed focuses on domain generalization (DG) and lacks implementation for the Unsupervised Domain Adaptation (UDA). Since we’re focusing on a UDA setting, we had to make major changes to the whole ML pipeline in DomainBed: data splits and preprocessing, algorithms, hyperparam tuning (because it’s different between DG and UDA) and eval.
> 2. Algorithms: DomainRobust implements 7 meta algorithms (11 variants) from scratch–these include adversarial training, TRADES, ARTUDA, SroUDA, and our proposed DART. These algorithms have important differences when it comes to the specifics of adversarial training, so we had to write a separate adversarial training code for each of the algorithms.
>
> Please let us know if you have more questions or concerns.

---

> > ### Author Response · Authors · 2023-11-21
> > **Follow-up on rebuttal**
> >
> > Dear Reviewer,
> >
> > Thank you again for your valuable comments! We have carefully addressed your comments.
> >
> > Please let us know if our replies have satisfactorily addressed your concerns. Since the discussion period will be closed soon, please do not hesitate to let us know if you have any further questions or if you require any additional clarification. Thank you in advance!

---

### Meta-Review · Area_Chair_hKru · 2023-12-08

**Metareview:**

This work tries to improve the adversarial robustness of UDA methods. Theoretically, they show that the target adversarial error can be bounded by the source error and the discrepancy. The experiments show that their methods outperform baselines. The idea is interesting, but all the reviewers think there are many limitations of this paper. For example, a gap between the theory and the algorithm is significantly; there are some issuses in the proof; the experimental significance of the proposed method is limited.

**Justification For Why Not Higher Score:**

N/A

**Justification For Why Not Lower Score:**

N/A

---

### Decision · Program_Chairs · 2024-01-16

Reject